# A standardized gnotobiotic mouse model harboring a minimal 15-member mouse gut microbiota recapitulates SOPF/SPF phenotypes

Marion Darnaud [1✉], Filipe De Vadder [2], Pascaline Bogeat[1], Lilia Boucinha[1], Anne-Laure Bulteau[2], Andrei Bunescu [1], Céline Couturier [1], Ana Delgado [1], Hélène Dugua[1], Céline Elie[1], Alban Mathieu[1], Tereza Novotná[3], Djomangan Adama Ouattara[1], Séverine Planel[1], Adrien Saliou[1], Dagmar Šrůtková[3], Jennifer Yansouni[1], Bärbel Stecher[4,5], Martin Schwarzer [3,6], François Leulier [1,2,6] & Andrea Tamellini [1,6]

*Mus musculus* is the classic mammalian model for biomedical research. Despite global efforts to standardize breeding and experimental procedures, the undefined composition and interindividual diversity of the microbiota of laboratory mice remains a limitation. In an attempt to standardize the gut microbiome in preclinical mouse studies, here we report the development of a simplified mouse microbiota composed of 15 strains from 7 of the 20 most prevalent bacterial families representative of the fecal microbiota of C57BL/6J Specific (and Opportunistic) Pathogen-Free (SPF/SOPF) animals and the derivation of a standardized gnotobiotic mouse model called GM15. GM15 recapitulates extensively the functionalities found in the C57BL/6J SOPF microbiota metagenome, and GM15 animals are phenotypically similar to SOPF or SPF animals in two different facilities. They are also less sensitive to the deleterious effects of post-weaning malnutrition. In this work, we show that the GM15 model provides increased reproducibility and robustness of preclinical studies by limiting the confounding effect of fluctuation in microbiota composition, and offers opportunities for research focused on how the microbiota shapes host physiology in health and disease.

[1] BIOASTER, Institut de Recherche Technologique, 40 avenue Tony Garnier, 69007 Lyon, France. [2] Institut de Génomique Fonctionnelle de Lyon, Université de Lyon, Ecole Normale Supérieure de Lyon, Centre National de la Recherche Scientifique, Université Claude Bernard Lyon 1, Unité Mixte de Recherche 5242, 46 Allée d'Italie, 69364, Lyon, Cedex 07, France. [3] Laboratory of Gnotobiology, Institute of Microbiology of the Czech Academy of Sciences, 54922 Nový Hrádek, Czech Republic. [4] Max von Pettenkofer Institute of Hygiene and Medical Microbiology, Ludwig-Maximilians-University of Munich, 80336 Munich, Germany. [5] German Center for Infection Research (DZIF), Partner Site, Munich, Germany. [6] These authors jointly supervised this work: Martin Schwarzer, François Leulier, Andrea Tamellini. ✉email: gnotobiology@bioaster.org

The intestinal microbiota is a complex and dynamic community largely composed of bacteria whose activity profoundly influences our health and diseases[1]. Advances in sequencing and analytical technologies coupled with improved computing tools have revolutionized the field of host–microbiota interaction[2]. These developments have provided an increased depth and accuracy in the study of intestinal microbial assemblages and activity for correlative studies with human health or disease traits. Despite these sophisticated descriptions of host–microbiome interaction phenomena, the underlying causal mechanisms remain largely elusive[3].

The use of model organisms plays a decisive role in the challenge to move from correlation to causal links in the host–microbiome field as they have long enabled researchers to identify the shared biological functions among living organisms, and facilitated the discovery of conserved molecular mechanisms governing the fundamental principles of biology[4]. Owing to its genetic and physiological similarities to humans, in addition to its rapid and prolific breeding, the mouse has been a classic mammalian model of choice for the past decades for biomedical research, and the host–microbiome field is no exception[5]. While the use of defined genetic backgrounds, as well as the absence of specific pathogens, is now a common practice in mouse studies[6], an important confounding factor is variability in the composition of the intestinal microbiota, which can cause marked phenotypical variations among experimental animals and animal facilities[7–9]. This parameter is under the influence of multiple elements such as genetics, diets, biological rhythms, and breeding conditions[10]. As a consequence, to restrain microbial diversity, efforts have been made to tailor protocols for microbiota-related mouse studies and to standardize mouse microbiota composition[10–14].

The mouse gut microbiota richness is usually estimated at more than 300 bacterial genera[5] and common inhabitants of the mouse intestine belong to seven bacterial phyla with *Firmicutes*, *Bacteroidetes*, and *Proteobacteria* being the most abundant ones[13,15,16]. The first attempt in standardizing the mouse microbial environment (initially to study immunocompromised mouse models) arose in the 1960s with the wide implementation of the specific pathogen-free (SPF) hygienic status of mouse husbandries[17]. Nowadays, SPF animals are obtained by re-deriving mouse strains by two-cell-stage embryo transfer to SPF recipients and subsequent postnatal inoculation with a cocktail of bacteria devoid of pathogens to homogenize microbial colonization within a given animal facility. SPF, and then specific (and opportunistic) pathogen-free (SOPF) inbred lines (lacking specific opportunist pathogens such as *Staphylococcus aureus* or *Pseudomonas aeruginosa*) now represents the common health standard for experimental mouse breeding[6]. However, despite the global efforts in standardizing the SPF procedures, the undefined nature and important interindividual diversity of the SPF microbiota remains a limitation in host–microbiome studies, since the scientific community still lacks a common SPF standard cocktail and rather use a facility-specific cocktail of bacteria[10]. Indeed, the microbiota fluctuates a lot with diet and environment, so it is impossible to have the exact same microbiota of SPF mice in two different facilities.

Microbial cultivation and gnotobiology offer attractive strategies to standardize the microbiota of mouse models. Germ-free (GF) animals (i.e., animals devoid of any living micro-organisms) are the originators of gnotobiotic animals (i.e., animals with a controlled microbiota) obtained by colonization with pure culture or cocktails of bacterial strains[18]. Recent efforts have been put into isolating, cultivating, and archiving isolated cultures of the dominant members of the mouse microbiota[19]. Gnotobiotic animals can be kept in isolators for several generations and offer the possibility of strict control of their microbial status.

Gnotobiotic models offering a different degree of microbial complexity have been developed in the past ranging from mono-colonization (monoxenic animals) to high diversity microbiota models such as conventionalized ex-GF animals using a donor microbiota[10,13,20]. Two models have emerged for breeding and long-term experimental purpose: the Altered Schaedler Flora (ASF) model and the recent Oligo-Mouse-Microbiota[12] (Oligo-MM[12]) model[21,22]. These models offer an enlarged microbial potential as compared to monoxenic mice while keeping the model simple and experimentally tractable as compared to conventionalized animals (SPF or SOPF).

The ASF was developed in the late 1970s by adding bacterial strains which better represented the microbiota of conventional mice to the initial Schaedler flora, a minimal microbial consortium that protected ex-GF mice from opportunistic pathogen colonization during breeding[23]. The ASF is composed of eight defined bacterial strains, which are stable over mouse generations. Most immune parameters are normalized in mice colonized with ASF when compared with SPF mice, but the fact that strains of the ASF model are not publicly available and that they are not all representative of the dominant members of the mouse microbiota remains an important limitation[10,13,21]. In addition, ASF mice differ substantially from SPF mice with respect to intestinal microbial biochemical activities and resistance to opportunistic pathogen colonization, probably owing to the limited phylogenic diversity and metabolic capabilities of the ASF consortium[10,13,24]. Recently, the Oligo-MM[12] model was developed[22]. It is a minimal microbiota gnotobiotic model composed of 12 defined cultivable mouse commensal bacteria from the miBC collection representing members of the major bacterial phyla of the mouse gut[19,25,26]. The community is transmissible and stable over consecutive mouse generations and animal facilities[27] and was recently used to reveal that the intestinal microbiota adapts to environmental changes by short-term effects of transcriptional reprogramming and adjustments in sub-strain proportions and long-term genomic positive selection[28]. Functionally, and unlike ASF, Oligo-MM[12] offers colonization resistance against *Salmonella enterica* serovar Typhimurium when supplemented with facultative anaerobic bacteria including *Escherichia coli*[22].

In an attempt to standardize preclinical studies in the host–microbiome field, we have developed a simplified mouse microbiota that is representative of SOPF microbiota at the functional level and derived a standardized gnotobiotic mouse model called GM15, which phenotypically mimics SOPF and SPF mice under standard dietary conditions in two different animal facilities. We demonstrate that under conditions of chronic physiological stress such as postweaning malnutrition on a low-protein diet, a dietary condition triggering stunting, GM15 microbiota shows improved capacities compared to a SOPF microbiota to buffer the deleterious effect of a depleted diet (DD) on mouse juvenile growth.

## Results

**In silico identification of the main bacterial families of a C57BL/6J SOPF fecal microbiota**. To define a minimal microbiota containing representative and prevalent bacteria from the gut of C57BL/6J SOPF mice, we analyzed the composition of fecal pellets from four C57BL/6J SOPF mice (obtained from Charles River Laboratories, France) by whole-genome sequencing (WGS) (two females and two males, which were littermates but housed in different cages from weaning at 3 weeks old, and feces were collected at 2 months old in our facility). An average of 13.4 million paired-end reads was obtained per sample with a length of 300 bp. The metagenomics data sets generated were classified using the Centrifuge software[29] and compared to the RefSeq

complete genome database[30], and 20 dominant families consistently present in all mice were identified (Fig. 1a). The profiling of metagenomic sequencing data pointed out a comparable distribution of bacterial families among the four tested C57BL/6J SOPF mice (Supplementary Fig. 1a, b). Moreover, bacterial species identification was possible for genome sequences with good phylogenetic resolution and already referenced in taxonomy databases. Interestingly, among the identified species, *Bacteroides acidifaciens*, *Clostridium cocleatum*, *Lactobacillus johnsonii*, *Ligilactobacillus murinus*, and *Limosilactobacillus reuteri* were previously identified as mouse-enriched and dominant intestinal bacteria[19], *Ligilactobacillus murinus*, *Parabacteroides goldsteinii*, and *Clostridium* strains are part of the ASF model[21], and strains of *Enterocloster clostridioformis*, *Limosilactobacillus reuteri* and *Bacteroides caecimuris* are part of the Oligo-MM[12] model[22].

**Isolation and taxonomic characterization of the GM15 bacterial strains**. We established four different strategies in order to isolate and culture a maximal number of representative strains of the 20 dominant bacterial families identified by our metagenomic sequencing analysis (Fig. 1b). First, we isolated the most prevalent strains from fecal pellets of C57BL/6J SOPF mice using non-selective agar media. Then, we used antibiotic selection to isolate resistant strains. We also used rumen enrichment to isolate strains from cecal content. Finally, we used fecal pellets of ASF mice to isolate additional strains. We obtained a collection of approximately 400 cultivable bacterial isolates. All isolates were prescreened by matrix-assisted laser desorption ionization time-of-flight mass spectrometry (MALDI-TOF MS) for dereplication prior to the first taxonomic identification by 16S ribosomal RNA (rRNA) gene Sanger sequencing. We selected 11 strains covering seven of the most representative and prevalent families of the intestinal microbiota of C57BL/6J SOPF mice and obtained four additional strains from the DSMZ miBC collection[19] to establish the GM15 consortium that covers most of the dominant bacterial families found in C57BL/6J SOPF animals (Fig. 1a, b). Actually, at the family level, the GM15 consortium putatively covers 63% of the SOPF consortium, which is more than the putative coverage previously available with ASF and Oligo-MM[12] models covering, respectively, 58% and 48%. In summary, GM15 is composed of two strains of *Bacteroidaceae*, one strain of *Tannerellaceae*, six strains of *Lachnospiraceae*, three strains of *Lactobacillaceae*, one strain of *Erysipelotrichaceae*, one strain of *Ruminococcaceae*, and one strain of *Enterobacteriaceae* (Supplementary Fig. 1c). We recently reported the draft genomes of the 15 strains and alignment against the NCBI database[31] allowed the identification of 12 strains at the species level and 3 strains at the family level[32], which are described as novel taxa in this study.

**In silico functional metagenomics analysis of the GM15 strains**. To gain insights into the functionalities encoded in the individual genomes of the GM15 members, the coding sequences of the 15 strains were converted into their respective protein sequences, which were annotated for clustering into KO (KEGG Orthology) groups. By merging the 15 assembled individual genomes, we found that the GM15 metagenome possesses 3890 nonredundant KO groups covering 44% of all protein-coding sequences. Besides, all GM15 strains possessed 3 to 64 unique functions, although *E. coli* Mt1B1 exhibited a vast repertoire of unique KO groups (10³), indicating that *E. coli* Mt1B1 is responsible for the GM15's functional metagenomic profile at 33%, while the 14 other strains contribute all together at 8% and the 59% remaining are associated to nonunique KO groups (Fig. 1c). It is noteworthy that KEGG module analysis is biased towards gene sets, pathways, and functional groups of well-characterized bacteria such as *E. coli*, which is by far the most studied bacterial species to date.

Next, we highlighted the in silico functionalities of the 15 selected strains associated with known enzymatic activities in the gut[33] (Fig. 1d). Again *E. coli* Mt1B1 is a major contributor, but each functionality is also covered by other strains at equivalent or lower levels. As expected, the enzymatic activities in the gut are correlated with the phylogenetic membership of the strains. For example, lactobacilli, which are a major part of the lactic acid bacteria group, the principal contributor to lactate dehydrogenase[34], clustered together. In addition, *Lachnospiraceae* are clustered with *Ruminococcaeae* and *Erysipelotrichaceae*, which include mainly bacteria with sporulation capabilities[35–37]. Finally, *Bacteroides* are clustered with *Parabacteroides*, whose species are predominant in the colonic mucus barrier and promote mucinase activity[38], and are significant producers of succinate, a major metabolic by-product[39,40]. Thus, different strains of a bacteria family and of other closely related bacterial families are capable of the same enzymatic activities in the gut. This can be essential for the generation of simplified nonspecific gnotobiotic models.

Then, we determined the functional coverage of the GM15 metagenome (i.e., the sum of the genomes of the 15 strains) relative to the KEGG modules of C57BL/6J SOPF mouse microbiota (covering 47% of all protein-encoding sequences) found in our initial metagenomic analysis (Fig. 1a). In addition, the KEGG modules from the Oligo-MM[12] and ASF microbiota (covering, respectively, 46% and 43% protein-coding sequences) were included for comparative analysis, as these consortia were previously used to generate gnotobiotic mice with a stable and defined mouse-derived microbiota[25,41]. The presence and completeness of KEGG modules were determined for each metagenome and used for hierarchical clustering (Fig. 1e). One cluster contained highly conserved modules in all mouse models (Fig. 1e, cluster 4 and Supplementary Data 1), including 140 pathways. We also identified clusters of modules that were not represented among the ASF and Oligo-MM[12] consortia, but specifically common to GM15 and SOPF (Fig. 1e, clusters 9 and 6 and Supplementary Data 1), comprising a total of 155 pathways indicating that qualitatively the GM15 metagenome covers functionalities found in SOPF microbiota that was lacking in ASF and Oligo-MM[12] models. Quantitatively, the defined consortia of GM15, Oligo-MM[12], and ASF covered, respectively, 72%, 54%, and 48% of the KEGG modules of the C57BL/6J SOPF microbiome, suggesting a superior functional potential of the GM15 community as compared to Oligo-MM[12] and ASF models. Thus, taken collectively, our in silico analysis suggests that the GM15 community carries a significant potential for enzymatic activities in the gut and recapitulates widely the functionalities found in C57BL/6J SOPF murine metagenome.

**Monitoring and stability assessment of GM15 gut microbiota**. To explore in vivo the functional potential of the GM15 community, we used our SOPF colony to produce GF mice and next-generated GM15 gnotobiotic animals. First, we investigated whether the GM15 consortium can stably colonize the mouse intestine over several generations. To this end, we developed a strain-specific quantitative polymerase chain reaction (qPCR) microfluidic assay, which allows simultaneous absolute quantification of the 15 strains, along with the global bacterial load in a given biological sample (Supplementary Table 1). Fecal samples from GF and SOPF mice were used as negative and positive controls. Only *Enterocloster clostridioformis* YL32, which was obtained from the DSMZ miBC collection, was not detected in

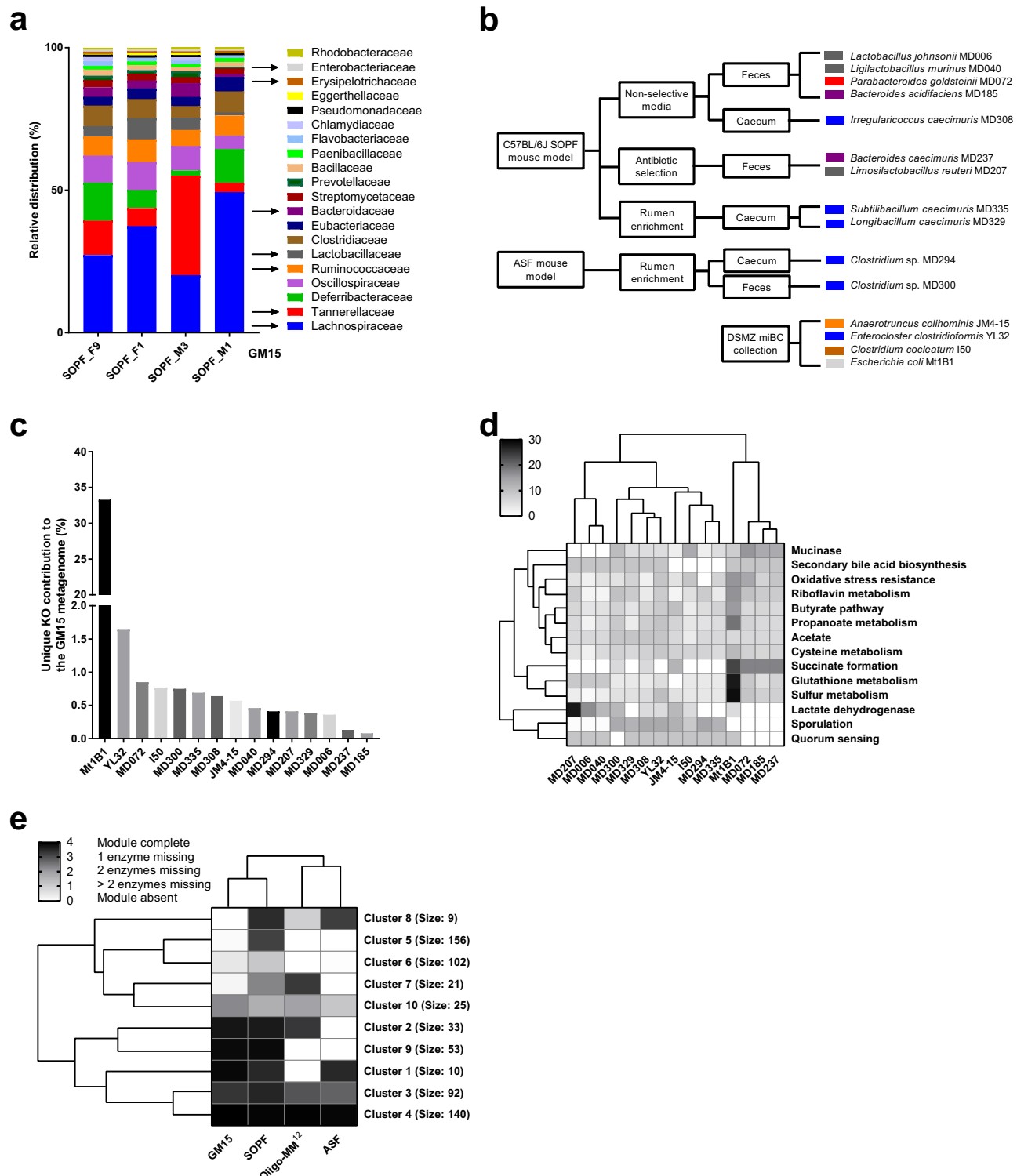

**Fig. 1 Selection, isolation, and functional analysis of GM15 strains. a** Representative and prevalent bacterial families identified from fecal samples of 4 C57BL/6J mice of our colony. **b** Eleven bacterial strains were isolated from fecal or cecal samples of either C57BL/6J or ASF mice using various culture methods. Four additional strains were obtained from the DSMZ collection. **c** Unique KO (KEGG Orthology) groups are those present in only one GM15 member. The unique contribution of each GM15 member is relatively evenly distributed (<2%), except *E. coli* whose unique functions contribute for 33%, although nonunique KO groups are primarily responsible for the GM15's metagenomic profile with 59% of the contribution. **d** Heatmap of hierarchical clustering of enzymatic activities in the gut for the 15 strains of the GM15 model. In quorum sensing (QS) function, only effector proteins were screened (because receptors suffer from similarities with nonspecific QS receptors). **e** Heatmap of hierarchical clustering of KEGG module distribution in the metagenomes of GM15, SOPF, ASF, and Oligo-MM[12] models. Clusters of KEGG modules are highlighted with their respective size (number of KEGG modules). The list of KEGG modules and clusters is shown in Supplementary Data 1. Source data are provided as a Source Data file.

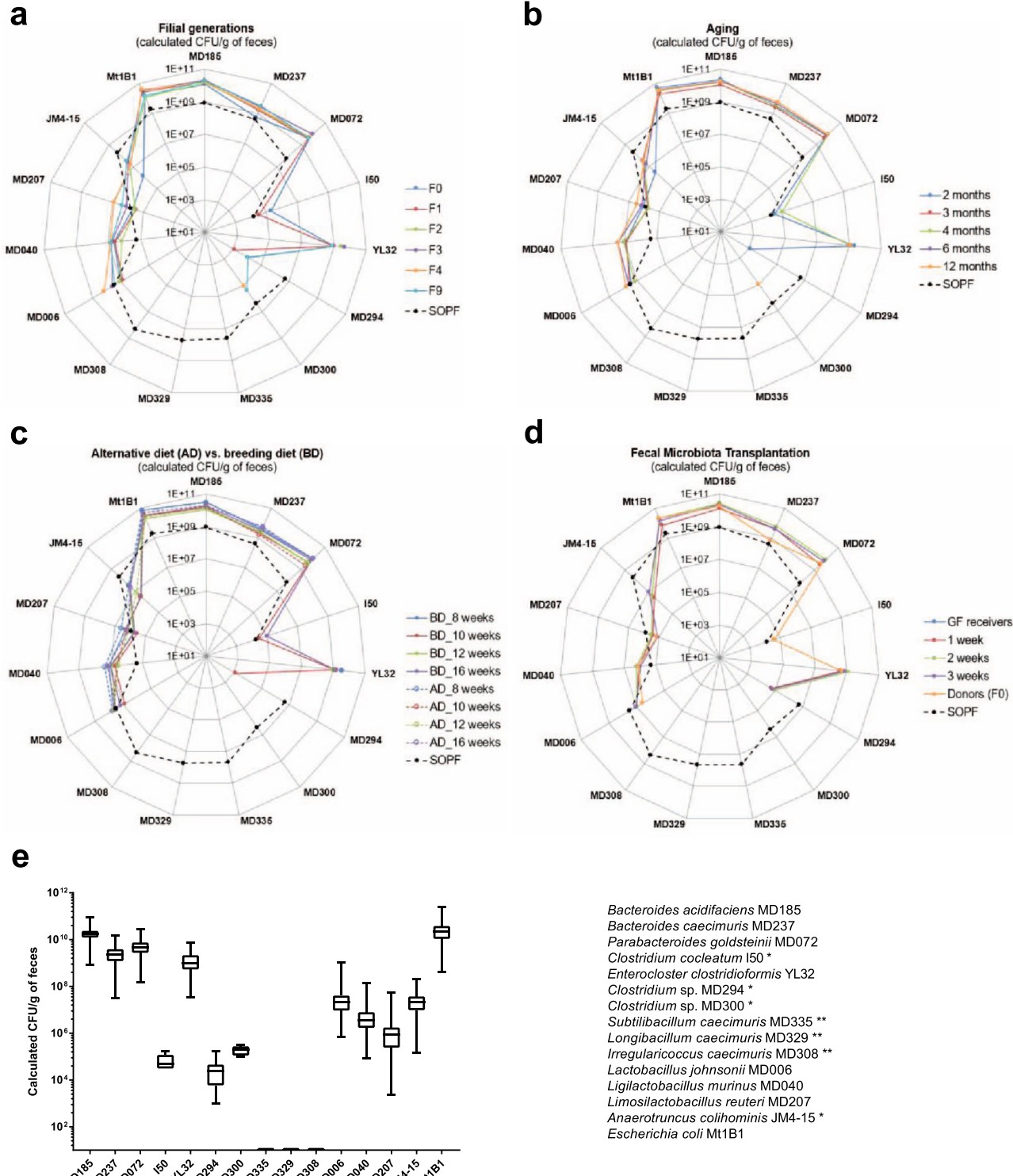

*Bacteroides acidifaciens* MD185
*Bacteroides caecimuris* MD237
*Parabacteroides goldsteinii* MD072
*Clostridium cocleatum* I50 *
*Enterocloster clostridioformis* YL32
*Clostridium* sp. MD294 *
*Clostridium* sp. MD300 *
*Subtilibacillum caecimuris* MD335 **
*Longibacillum caecimuris* MD329 **
*Irregularicoccus caecimuris* MD308 **
*Lactobacillus johnsonii* MD006
*Ligilactobacillus murinus* MD040
*Limosilactobacillus reuteri* MD207
*Anaerotruncus colihominis* JM4-15 *
*Escherichia coli* Mt1B1

our SOPF mice. Co-monitoring of specific and total bacteria aimed to detect any bacterial load imbalance caused by contamination in gnotobiotic isolators.

Eight-week-old GF mice (five breeding pairs; GM15 founders, F0; Supplementary Fig. 2a) were inoculated by oral gavage with a fresh frozen mixture of the 15 strains and bred in sterilized positive pressure isolators up to the F9 filial generation. All strains except *Subtilibacillum caecimuris* MD335, *Longibacillum caecimuris* MD329, and *Irregularicoccus caecimuris* MD308 were above the detection limit of our qPCR microfluidic assay in the

fecal samples of most individual mice (GM15 founders and progenies from the nine consecutive generations; Fig. 2a and Supplementary Fig. 3a). *Anaerotruncus colihominis* JM4-15, *Clostridium* sp. MD294, *Clostridium* sp. MD300, and *Clostridium cocleatum* I50 were occasionally below the detection limit, but the detection of the remaining strains was reproducible between fecal and cecal samples from individual mice in the second filial generation (Fig. 2a and Supplementary Fig. 3b). Taken together, these results indicate stable colonization and effective vertical transmission of at least 12 strains out of the 15 inoculated. Based

**Fig. 2 Stability assessments of the GM15 mice gut microbiota over filial generations, in aging, under diet change, and through FMT. a–d** SOPF groups show the distribution of each GM15 strain in the complex gut microbiota of 8-week-old SOPF mice. The absolute quantification of each strain was determined by specific qPCR microfluidic assay. *Strains I50, MD294, MD300, and JM4-15 were at the detection limit of the qPCR microfluidic assay and thus were not detected in all samples. **Strains MD335, MD329, and MD308 were below the detection limit of the qPCR microfluidic assay. Strain YL32, obtained from the DSMZ collection, was not detected in our SOPF colony. **a** Radar plot showing the GM15 strains distribution in feces of C57BL/6J GF mice colonized with the GM15 community (F0, $n = 10$) and bred for consecutive generations (F1–F9, $n = 22$–8). **b** Radar plot showing the overall stability of the GM15 community composition in feces collected from nine mice between 2 and 12 months of age (two mice died at 12 months of age). **c** Radar plot showing that an alternative diet, such as a maintenance diet, can be used for 4 weeks, and then reversed to the breeding diet for 4 more weeks, without modifying the composition of the gut microbiota of GM15 mice ($n = 18$) compared to mice fed all along with the breeding diet ($n = 9$). **d** Radar plot showing the feasibility of GM15 fecal microbiota transplantation to GF mice ($n = 9$). **e** Box plots showing the low variability of the GM15 strains concentrations considering all fecal samples of mice from generation F1 to F9 ($n = 113$), from generation F1 at 12 months old ($n = 7$), from generation F1 under diet change ($n = 18$ per time point 8, 10, 12, and 16 weeks) and control breeding diet ($n = 9$ per time point 8, 10, 12, and 16 weeks), and fecal samples collected 3 weeks post fecal microbiota transplantation in GF mice ($n = 9$). Box plots extend from the 25th to 75th percentiles and show the center line as the median. Whiskers represent min and max data. Source data are provided as a Source Data file.

on the results of our qPCR assay, we consider that the three remaining strains: *Subtilibacillum caecimuris* MD335, *Longibacillum caecimuris* MD329, and *Irregularicoccus caecimuris* MD308, either did not efficiently colonize the animals, live in the cecum or the colon below the detection limit of our qPCR assay, or predominantly live in other gastrointestinal niches than those sampled.

Next, we evaluated the effect of aging on the GM15 community by following individual mice of the first filial generation between 2 and 12 months of age (Supplementary Fig. 2a). Overall, no changes in the qualitative and quantitative composition of the GM15 consortium were detected (Fig. 2b). Then, we asked if the composition of the GM15 community was modulated by substituting the breeding diet (BD) with an alternative maintenance diet. This alternative diet is quasi isocaloric but its nutritional composition differs by 1.3- and 1.6-fold fewer proteins and lipids, respectively, and 1.2-fold more carbohydrates than the BD. An alternative diet was administered to 8-week-old GM15 mice for 4 weeks. We collected fecal samples before diet change, after 2 and 4 weeks, and again after 4 weeks back to the BD (Supplementary Fig. 2a). We did not detect any significant changes in the GM15 composition under these conditions (Fig. 2c). In addition, successful fecal microbiota transplantation (FMT) from GM15 founders to GF mice was confirmed by strain-specific qPCR of the ex-GF mice feces collected at weeks 1, 2, and 3 posttransplantation (Supplementary Fig. 2a and Fig. 2d).

When all the data are analyzed together, we notice a limited fluctuation (maximum 2 Log 10-fold change) of each member of the GM15 gut microbiota's load among the conditions tested (Fig. 2e). Therefore, we conclude that the GM15 community is stable upon adult colonization, among filial generations, during aging, upon mild dietary fluctuations, and can be transmitted efficiently by FMT.

**The GM15 microbial community recapitulates SOPF macroscopic phenotype**. GF and published gnotobiotic mice display anatomical alterations compared to SOPF mice, such as enlarged cecum, along with physiological and metabolic differences[22,42]. To phenotypically assess the gnotobiotic GM15 model, we designed a comparative study to evaluate the steady-state macroscopic, immune, metabolic, and endocrine phenotypes of GF, SOPF, and GM15 mice. In addition, all phenotyping was achieved across two generations, F1 and F2, to strengthen data analysis.

Initially, we evaluated the reproduction performance of the GM15 model by recording the period from mating to offspring delivery (Fig. 3a), the number of pups per litter (Fig. 3b), and the perinatal mortality (Fig. 3c). GM15 mice behaved like SOPF mice, with the exception of one less progeny per mean litter. Indeed, the distribution of progeny per litter in GM15 mice ranged more

evenly from 3 to 9 pups compared to SOPF mice, whose mean number of pups per litter was mostly centered on 7 or 8 pups. Then, we quantified the food intake relative to body weight after weaning at 4 weeks of age and observed no significant difference between the three groups despite a marked increased variation among GF animals, which is not detected in GM15 and SOPF animals (Fig. 3d). Next, we studied postnatal growth parameters. Male and female GM15 animals gained weight (Fig. 3e) and size (Fig. 3f) like SOPF mice, although the growth curves from the two sexes differ. We then studied internal organ size. As expected, the characteristic cecum enlargement seen in GF animals was reduced in GM15 mice (Fig. 3g). The weights of GM15 and SOPF brain, liver, and spleen were equivalent and larger than those of GF mice (Fig. 3g). The bone size was also identical in GM15 and SOPF mice and larger than that of GF mice (Fig. 3h). Taken together, these results confirm that the gut microbiota contributes to somatic tissue growth[43,44] and that the GM15 simplified microbiota is sufficient to largely recapitulate the breeding and growth performance of SOPF mice by compensating the physiological limitations of GF mice.

**The GM15 simplified microbiota partially restores SOPF immune phenotype**. It is now well established that host-specific bacteria consortia influence intestinal and systemic immune maturation[45,46]. We thus profiled the basal immune parameters of GM15 animals and compared them to SOPF and GF animals at 7–8 weeks of age. For that purpose, we analyzed immunoglobulin (Ig) levels in feces and sera and cytokine levels in the serum. In parallel, we assessed viable leukocytes by measuring CD45+ cell count and analyzed T cells, B cells, natural killer (NK) cells, and monocytes or dendritic cell (DC) population frequencies in whole blood and several lymphoid organs (spleen, thymus, Peyer's patches (PPs) and mesenteric lymph nodes (MLNs)).

Among already described immunodeficiencies in GF mice[47,48], we observed that the GM15 community restored the production of IgA in the gut and in serum, and IgG2b in the serum, at levels equivalent to those detected in SOPF mice (Fig. 4a–c). Notably, the highest levels of circulating IgA were not correlated with the highest levels of fecal IgA, and intragroup variability was not related to gut microbiota composition, which was homogeneous between individuals. In addition, circulating interleukin-22 (IL-22) levels, which is one of the key intestinal cytokines, were restored as well in GM15 mice (Fig. 4d). Furthermore, it has been previously shown that isolated lymphoid structures such as PPs and associated cellularity were strongly increased with microbiota diversity, whereas the total cell numbers in MLNs were comparable between GF and SPF mice[49,50]. As described in the literature, the number of PPs collected was higher in SOPF mice compared to GF and partially restored in GM15 mice (Fig. 4e).

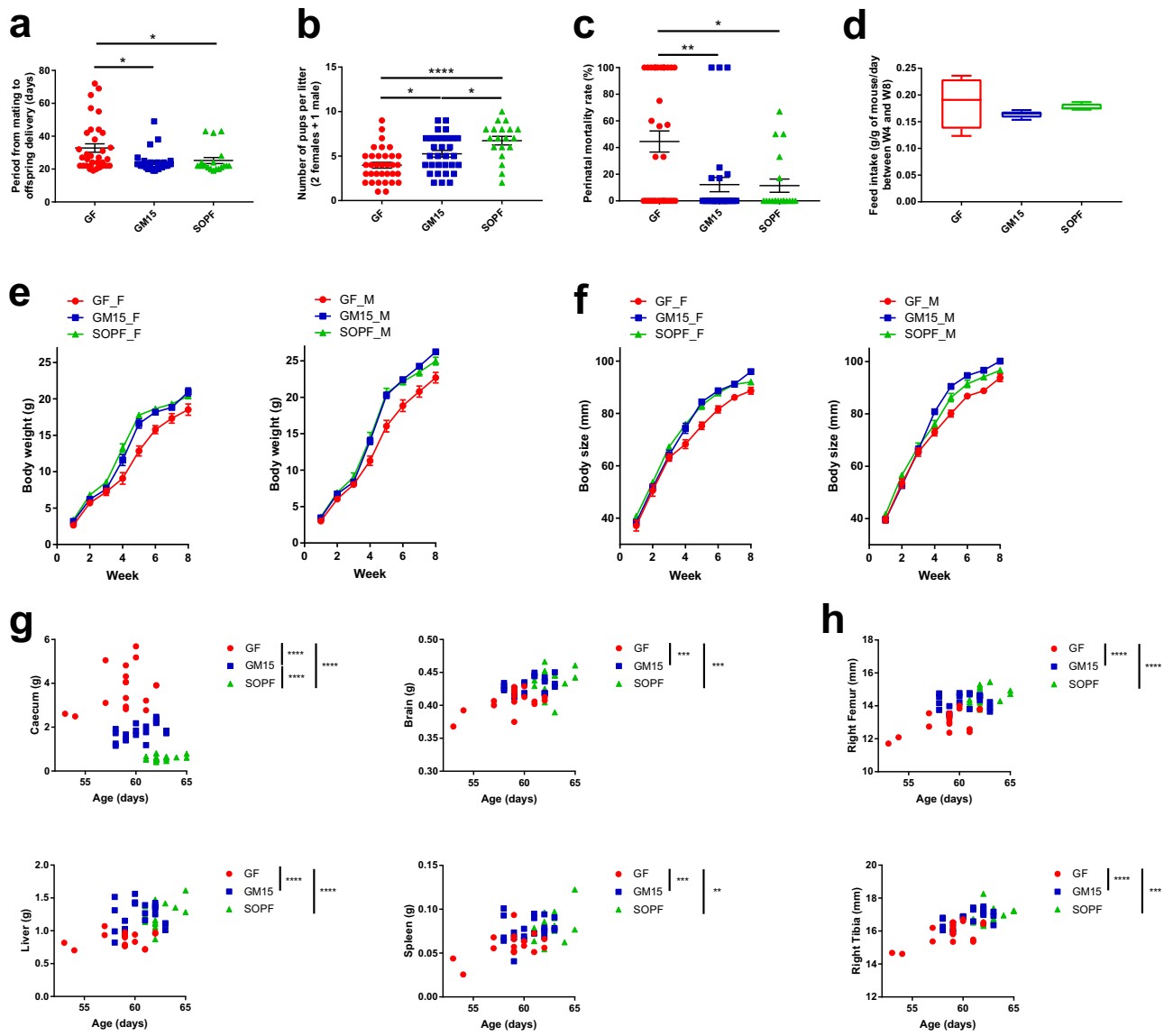

**Fig. 3 Macroscopic phenotyping of mice from two consecutive filial generations (F1–F2). a–c** One-way ANOVA analyses of the reproductive performance, where dots, lines, and error bars represent, respectively, individual litters (34 GF, 31 GM15, and 19 SOPF), means, and SEM. **a** Pups delivery. Data are represented as days after mating. Dunn's multiple comparison analyses. **b** Number of pups per litter. Data are represented as an individual number. Tukey's multiple comparison analysis. **c** Perinatal mortality. Data are represented as percentages of mortality. Dunn's multiple comparison analyses. **d** Feed intake normalized per gram of mouse per day over 4 weeks (17 GF, 21 GM15, and 20 SOPF mice housed in 6, 5, and 4 cages, respectively). Box plots extend from the 25th to 75th percentiles and show the center line as the median. Whiskers represent min and max data. **e, f** Body growth of mice bred with their mothers until week 4, where lines and error bars represent, respectively, means and SEM. Female (F) and male (M), respectively. Body weight and size curves. GF mice (7 F, 10 M), GM15 mice (9 F, 12 M), and SOPF mice (10 F, 10 M). **g, h** Organs weight or size impacted by age, filial generation, or sex were analyzed by the *F* test for multiple linear regressions, otherwise by one-way ANOVA. Each animal is represented by a dot at the age of the sacrifice (17 GF, 21 GM15, and 20 SOPF). *$P < 0.05$, **$P < 0.01$, ***$P < 0.001$, and ****$P < 0.0001$. Source data are provided as a Source Data file.

No or minor differences were observed between the three groups for CD45+ cell count (Fig. 4f and Supplementary Fig. 4a, b). A slight increase was observed in CD45+ cell count in SOPF compared to GF mice with apparent intermediate CD45+ cell count in GM15 mice in the spleen, thymus, and PPs and with no difference in MLNs (Fig. 4g–i and Supplementary Fig. 4a, b). In 2018, Kennedy et al. published an overview of the literature describing the main immune cell populations modifications observed in GF mice in different organs[51]. Interestingly, the great majority of populations impacted showed similar frequencies in GM15 mice and SOPF compared to GF mice (Supplementary Figs. 4a–f and 5a–e). Monocytes in whole blood, CD4+ T cells in the thymus, CD8+ T cells in MLNs, DCs in PPs, B cells in MLNs,

and whole blood and NK cells in the spleen were restored in GM15 mice. Among those cell populations, only B cells in GF whole blood showed higher concentrations compared to those described in the literature[46]. Our observations of NK cell decrease in the spleen and DC increase in PPs in GF mice compared to SOPF mice were not yet described.

Minor differences were observed between GM15 mice and SOPF, but not with GF mice for DC in the spleen and CD4+ T cells in PPs, whereas both populations were described to be decreased in GF mice (Supplementary Figs. 4a–e and 5b, e). Only NK cells in PPs showed similar and increased frequencies in GM15 and GF mice compared to SOPF mice (Supplementary Figs. 4a–e and 5e). In the same way, minor differences were

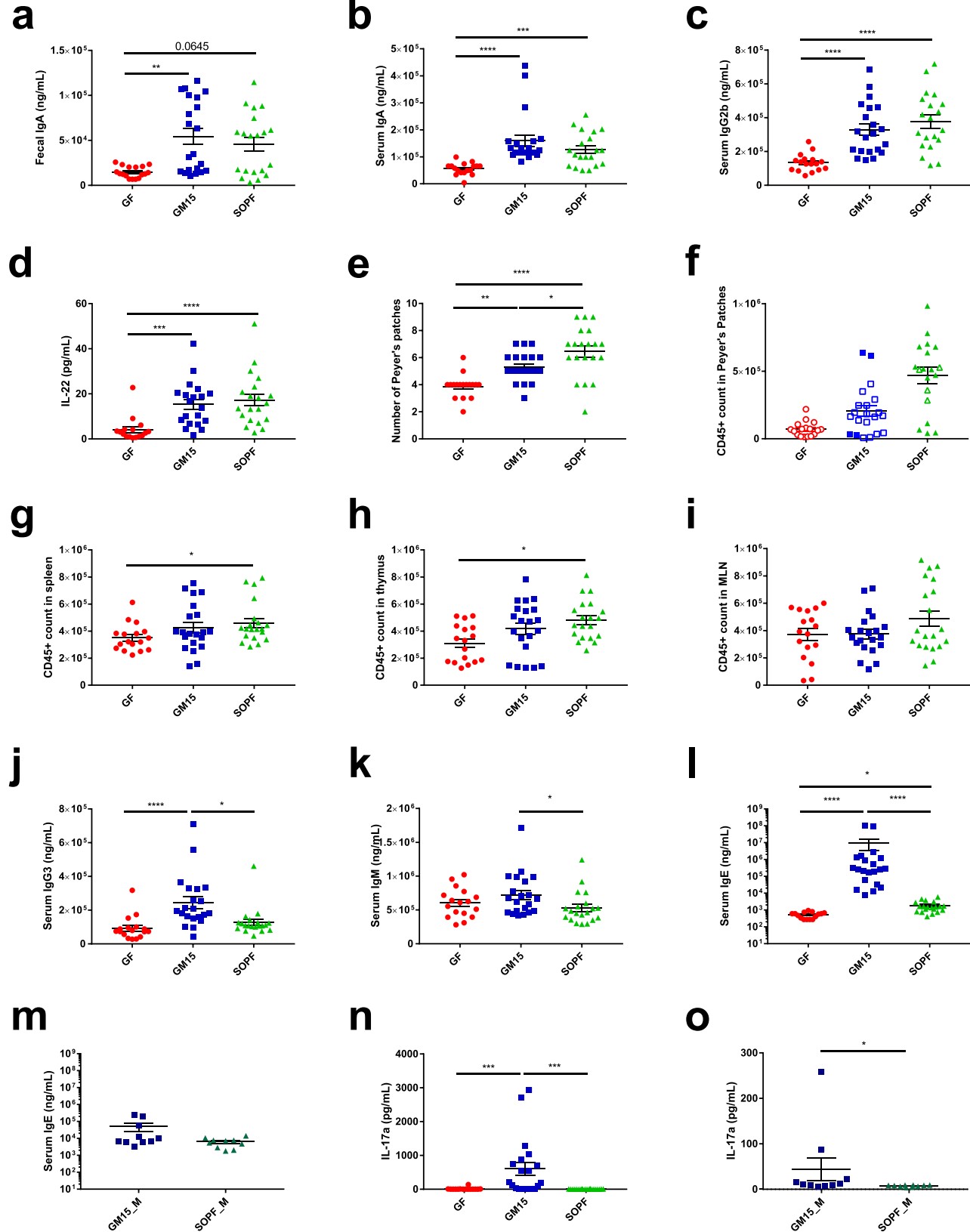

observed as well for some serum Ig levels. In our model, IgG3 levels were increased in GM15 mice compared to GF and SOPF mice and IgM levels were slightly increased in GM15 mice compared to SOPF (Fig. 4j, k). Of note, those Ig levels were lower in our SOPF mice than levels already described in SPF mice[47,48].

Surprisingly, the first set of GM15 mice (F1 and F2) born to a colonized mother presented elevated levels of IgE in serum compared to GF and SOPF mice (Fig. 4l). Such IgE levels may result from parasitic infections, immunodeficiencies, or long-lived IgE-producing plasma cells generated by food antigens in mice

**Fig. 4 Immune phenotyping through serum and fecal immunoglobulin subtyping, circulating cytokines analysis, and immune cell populations analysis in different organs from mice from two consecutive filial generations (F1–F2). a**–**o** Dot plots where dots, lines, and error bars represent, respectively, individual mice, means, and SEM. **a**–**c** Fecal IgA, serum IgA, and IgG2b Luminex analysis. *F* test for multiple linear regression analysis (fecal IgA) and one-way ANOVA followed by Dunn's multiple comparison analyses (serum IgA, IgG2b) (17 GF, 21 GM15, and 20 SOPF). **d** Circulating IL-22 level Luminex analysis. One-way ANOVA followed by Dunn's multiple comparison analyses (16 GF, 20 GM15, and 20 SOPF). **e** PP number. One-way ANOVA followed by Tukey's multiple comparison analysis (17 GF, 21 GM15, and 20 SOPF). **f** CD45+ cell count in PPs. CD45+ cell count per million of viable cells stained (filled symbol) or CD45+ cell count per total viable cells stained when <1 M cells were isolated (empty symbol). No statistical test (17 GF, 21 GM15, and 19 SOPF). **g**–**i** CD45+ cell count comparison in the spleen, thymus, and MLNs by flow cytometry (CD45+ count per million of viable cells stained). One-way ANOVA followed by Dunn's multiple comparison analyses (17 GF, 21 GM15, and 20 SOPF). **j**, **k** IgG3 and IgM Luminex analysis. One-way ANOVA followed by Dunn's multiple comparison analyses (17 GF, 21 GM15, and 20 SOPF). **l**, **m** IgE Luminex analysis. One-way ANOVA followed by Dunn's multiple comparison analyses and Mann–Whitney test, respectively (15 GF, 21 GM15, and 20 SOPF; additional F5–F6 male mice: 10 GM15_M and 10 SOPF_M). **n**, **o** Circulating IL-17a level Luminex analysis. One-way ANOVA followed by Dunn's multiple comparison analyses and Mann–Whitney test, respectively (16 GF, 20 GM15, and 20 SOPF; additional F5–F6 male mice: 10 GM15_M and 10 SOPF_M). *$P < 0.05$, **$P < 0.01$, ***$P < 0.001$, and ****$P < 0.0001$. Source data are provided as a Source Data file.

with low-diversity microbiota during early life[47,52,53]. Of note, unlike the previous study, our GF mice did not show increased IgE levels compared to SOPF[47]. Independent tests following FELASA guidelines (CR EU RADS, France) rejected the infection hypothesis of the GM15 mice, which were negative for ectoparasites and endoparasites, respiratory- and intestinal-specific pathogenic bacteria, and infectious agents, as well as viruses. To confirm this observation, we sampled again serum from additional GM15 animals (F5 and F6) several months after the initial test and quantified the IgE circulating levels with two independent methods. Although both Luminex and ELISA assays were calibrated against purified mouse IgEκ, they provided very different ranges of IgE values. However, both methods indicate that this second set of GM15 animals had normal levels of circulating IgE (Fig. 4m and Supplementary Fig. 5f). We also found that circulating IL-17a levels were increased in the first set of GM15 mice compared to SOPF and GF mice (Fig. 4n), but the IL-17a levels were much lower in the second set of GM15 animals and ultimately similar to SOPF mice, apart from one outlier (Fig. 4o).

Taken collectively, our results indicate that the GM15 community is sufficient to restore some key parameters of the immune response lacking in GF animals such as modified levels of serum and fecal IgA, serum IgG2b, serum IL-22, PP number, CD45+ cell count, and immune cell population frequencies in different organs close to that detected in SOPF animals.

**Low-complexity GM15 microbiota shares more metabolic traits with SOPF than GF mice.** The gut microbiota influences multiple host metabolic pathways by providing metabolites to its host and also shapes interorgan communication within the body by influencing the production and activity of endocrine signals[54,55]. One-dimensional proton nuclear magnetic resonance spectrometry ($^1$H NMR) was previously applied to investigate how the gut microbiota impacts host metabolism using mouse models[56,57], or human cohorts[58]. Using this technology, we analyzed the metabolic profile of plasma samples from GF, GM15, and SOPF mice and were able to quantify a total of 57 polar metabolites and 5 nonpolar metabolites. It could be seen that only a few of them were significantly affected by sex (Supplementary Data 2). A principal component analysis (PCA) based on quantified polar metabolites showed sequential alignment on the first principal component (9.2% of total variance) according to microbiota complexity of GF, GM15, and SOPF samples, independently of F1 and F2 generations (Fig. 5a). The metabolite composition of GF plasma resulted only from the host metabolic activity and represented a basal phenotype. Conversely, the SOPF mice harboring a very diverse gut microbiota, exhibited a larger

panel of metabolic activities based on host–bacterial and bacterial–bacterial interactions. In between, the GM15 low-complexity community contributed to a lower extent to the metabolic phenotype. We performed the equivalent statistical analysis using binned NMR spectra instead of quantified metabolites and as expected we obtained the same results (Supplementary Fig. 6). Next, we performed a discriminant analysis across all samples to highlight the specific metabolic signatures of each group. Nine metabolites emphasized significant variation, although it is noteworthy that the calculated distance placed GM15 closer to SOPF than to GF (Fig. 5b). Dimethylamine and isopropanol tended toward GF mice, but concentrations were low, and taken alone there was no difference between groups (Supplementary Data 2). As previously reported, GF mice had higher plasma levels of glycine[59] and reduced plasma acetate concentration[60]. On the contrary, SOPF mice harboring a diverse microbiota had higher plasma levels of acetate, dimethyl sulfone, 3-hydroxybutyrate, and propionate[54,61]. As for GM15 mice, less methanol and more citrate were detected. Interestingly, methanol may occur as a result of fermentation by gut bacteria and can stimulate citric acid fermentation[62,63]. Thus, the simplified gut microbiota of GM15 mice may produce less methanol and/or microbially produced methanol may be used to form citrate. These specific metabolic signatures may be used in the future as a panel of biomarkers to identify the GM15 model. Besides, the analysis of the plasma nonpolar metabolites indicated that GM15 colonization was sufficient to reduce free cholesterol and phosphatidylcholine as observed in SOPF mice (Fig. 5c). The three additional nonpolar metabolites detected were equivalent in all mice (Supplementary Data 2).

Finally, we investigated the circulating levels of key metabolic hormones and growth factors. To allow different types of analyses using the same blood sample, mice were not fasted. Most interestingly, as compared to levels detected in GF animals the GM15 gut microbiota was able to restore the levels of circulating insulin growth factor 1 (IGF-1) to the levels observed in SOPF animals. IGF-1 is an essential growth factor promoting systemic and tissue growth[64,65] and this observation correlates well with the improved macroscopic growth of GM15 mice compared to GF animals (Figs. 5d and 3e–h). Corticosterone levels, which are high in GF and low in SOPF animals, were also normalized by GM15 bacterial colonization[66] (Fig. 5e), indicating that the GM15 community seems as efficient as a complex SOPF microbiota at utilizing host metabolites and promoting steroidogenesis and growth factor production.

Collectively, our results reveal that GM15 animals stand out from GF mice and recapitulate many of the SOPF metabolic features even though some differences exist.

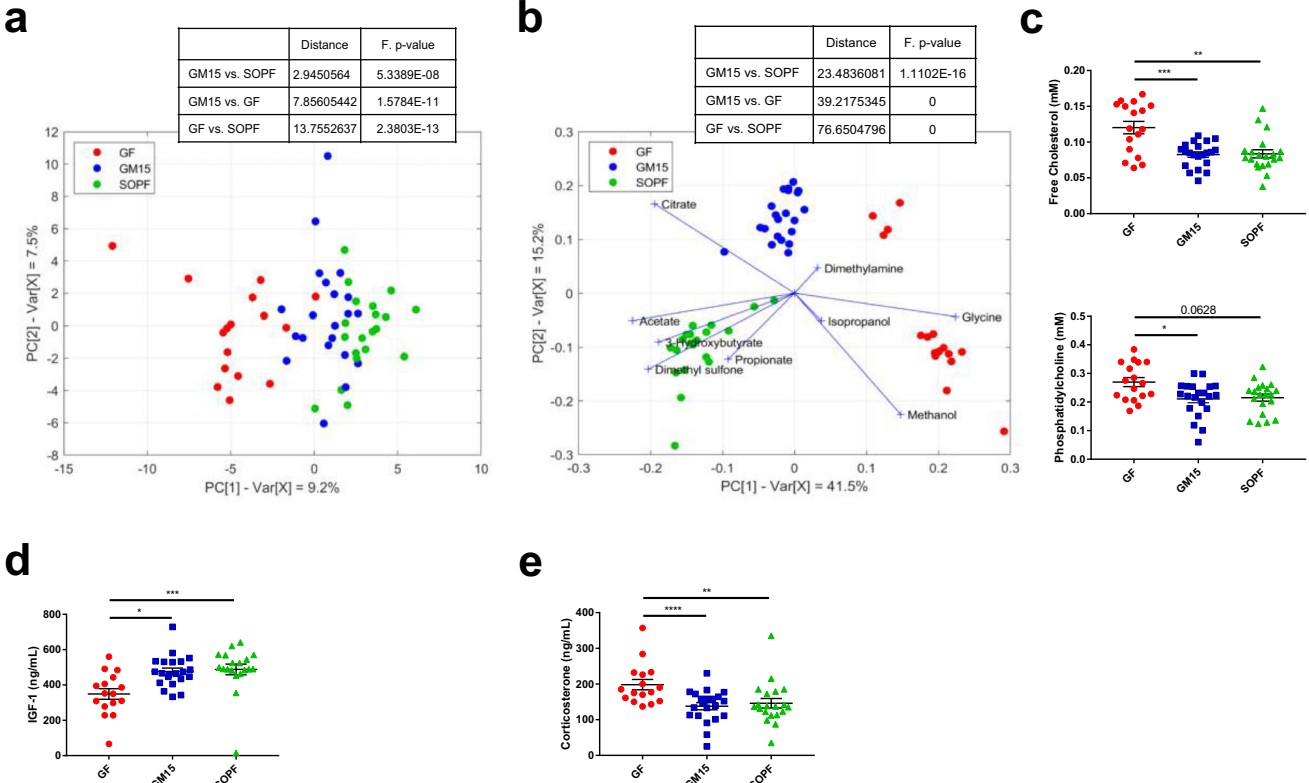

**Fig. 5 Metabolic phenotyping of mice from two consecutive filial generations (F1–F2). a** PCA score plot representing the distribution of the polar metabolite composition along the two first principal components (16 GF, 21 GM15, and 20 SOPF). **b** Biplot representing the projection of samples in the PLS-DA score plot given by the two first components and the projection of the contribution of significant polar metabolites (16 GF, 21 GM15, and 20 SOPF). **c–e** Dot plots where dots, lines, and error bars represent, respectively, individual mice, means, and SEM. **c** F test for multiple linear regressions representing free cholesterol and phosphatidylcholine (17 GF, 21 GM15, and 20 SOPF). **d** Serum IGF-1 level Luminex analysis. One-way ANOVA followed by Dunn's multiple comparison analysis (16 GF, 21 GM15, and 19 SOPF). **e** Serum corticosterone level Luminex analysis. F test for multiple linear regressions (16 GF, 21 GM15, and 20 SOPF). *$P < 0.05$, **$P < 0.01$, ***$P < 0.001$, and ****$P < 0.0001$. Source data are provided as a Source Data file.

**GM15 male mice are less sensitive to diet-induced stunting than SOPF mice.** Previous work by our lab and others[43,67,68] has shown that the gut microbiota influences pathogenesis associated with chronic undernutrition, particularly diet-induced stunting. Consequently, we sought to study our newly established gnotobiotic mouse model under severe nutritional stress, induced by a nutrient DD (containing 4% protein and 2% lipids) expected to trigger stunting. Based on our previous experience with the juvenile chronic undernutrition model, diet-induced stunting is more prominent in male mice compared to female mice (Supplementary Fig. 7). We thus fed male mice a BD or a DD from postnatal day 21 (i.e., the day of weaning) until postnatal day 56.

As shown in Fig. 6a, b and similarly to our previous observation (Fig. 3e, f), GM15 and SOPF mice grew well on the BD as they show similar body weight and size gains. Next, we confirmed that the DD triggered almost full stunting of both the juvenile GM15 and SOPF mice, characterized by the flattening of their weight and size curves (Fig. 6a, b). However, the GM15 mice performed better in terms of growth as both their body weight and size were significantly less impacted than SOPF animals by the DD. While the cecum of GM15 was enlarged (Fig. 6c and Supplementary Fig. 8a), this variation could not account for the total weight difference between GM15 and SOPF animals. We thus compared the sizes and weights of nine other organs (Fig. 6d–f and Supplementary Fig. 8b–g) in order to account for these variations and using two-way analysis of variance (ANOVA), we confirmed that diet was the main driver of the growth phenotype. However,

we found that the increase in body size observed in GM15 mice on the DD could be correlated to a significant increase in the size of the tibia (Fig. 6d). Although we did not find any other significant differences in other single organ sizes or weights between GM15 and SOPF animals on DD, we observed a clear tendency of an increase in GM15 compared to SOPF animals for most parameters tested (Fig. 6e, f and Supplementary Fig. 8b–g). We tested this tendency by integrating all the phenotypical parameters in a PCA. We first established which parameters were correlated to the phenotype (Supplementary Fig. 8h), thus excluding brown adipose tissue from the analysis as a noncorrelating parameter. The PCA revealed that, under BD, the GM15 phenotype was part of the spectrum of the SOPF phenotype (Fig. 6g). However, under DD, there is a clear shift between the GM15 and SOPF phenotypical space (Fig. 6g, dotted lines). Our results thus indicate that under nutritional stress, GM15 microbiota buffers diet-induced stunting slightly more effectively than a SOPF microbiota. As IGF-1 is an important driver of the diet and microbiota-mediated growth promotion[43], we assessed IGF-1 levels in animals, and while the levels massively drop in the DD conditions, we did not detect any difference between GM15 and SOPF mice at day 56 on the DD (Fig. 6h). However, we cannot exclude that slight variation in size between GM15 and SOPF animals on DD results from a differential secretion of IGF-1 before day 56, when the growth rate of the animal is maximal (Fig. 6a, b).

Taken together, our data show that animals bearing the simplified GM15 microbial community perform similarly when

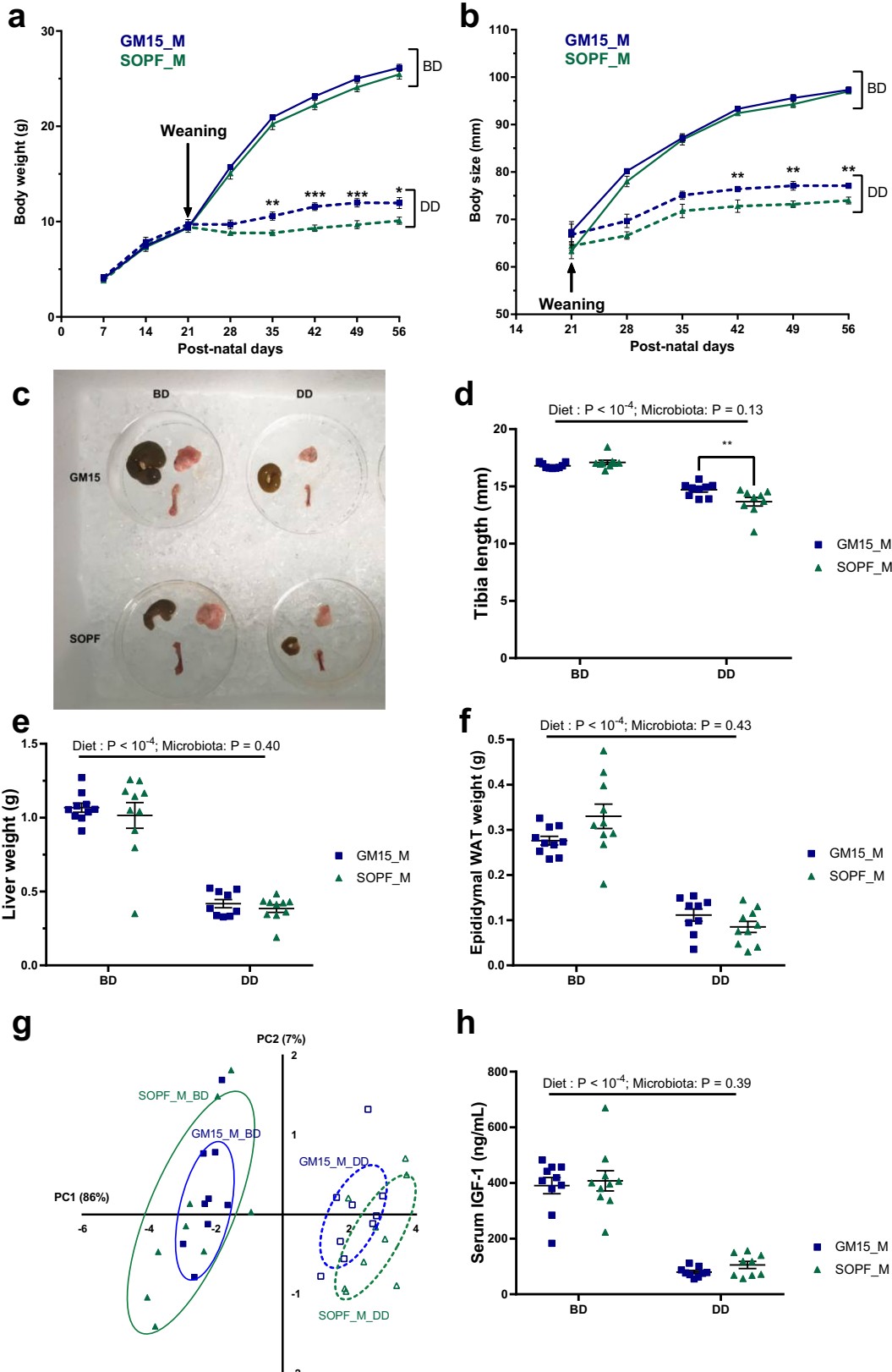

compared to animals bearing a complex SOPF microbiota under non-stressful conditions. However, the GM15 microbiota seems partially protective against the deleterious effects of chronic undernutrition, as compared to SOPF mice.

**The GM15 model growth and diet-induced stunting phenotypes are reproducible in another gnotobiotic facility.** Reproducibility is imperative to preclinical research, especially in the microbiome field. Therefore, to test the transferability and

**Fig. 6 GM15 mice and SOPF response under postweaning chronic undernutrition. a–h** Dot plots where dots, lines, and error bars represent respectively individual mice, means, and SEM. Male (M), breeding diet (BD), and depleted diet (DD), respectively. Body weight (**a**) and body size (**b**) were measured every week from weaning to postnatal day 56 (ten males per group). **c** Representative picture of the cecum, subcutaneous fat, and right femur of mice at day 56. Tibia length (nine males per group) (**d**), liver (ten males per group, except nine GM15_M_DD) (**e**), and epididymal white adipose tissue (WAT) (ten males per group, except nine GM15_M_DD) (**f**) weight were measured. **g** PCA of tissues weight and bones size under control conditions (BD) or under nutritional stress (DD). Brown adipose tissue was excluded, given its lack of correlation with other parameters (Supplementary Fig. 8h). Percentages on each axis indicate the variance associated with each coordinate (nine males per group). **h** Serum IGF-1 at day 56 (ten and nine males per BD and DD groups, respectively). *P* values after two-way ANOVA were adjusted for Sidak's post hoc test for multiple comparisons. *$P < 0.05$, **$P < 0.01$, and ***$P < 0.001$. Source data are provided as a Source Data file.

reproducibility of the GM15 model, we established it in a second gnotobiotic facility (Facility 2—Laboratory of Gnotobiology at Institute of Microbiology of the Czech Academy of Sciences, Novy Hradek, Czech Republic). We colonized GF animals with the GM15 consortia using the same protocol described for facility 1. Using qPCR microfluidic assays, we analyzed the bacterial titers in fecal samples of the GM15 animals and control SPF animals from the same facility, bred in the same condition as GM15 animals (Supplementary Fig. 2b). Similarly to GM15 colonized mice at facility 1, all strains except *Subtilibacillum caecimuris* MD335, *Longibacillum caecimuris* MD329, and *I. caecimuris* MD308 were above the detection limit in GM15 individual mice (Fig. 7a). GM15 strain levels in facilities 1 and 2 were very similar (Fig. 7b), demonstrating that the GM15 model can be effectively transferred and established in a reproducible manner in different gnotobiotic facilities.

We next compared the titers of each strain individually in GM15 and SOPF/SPF animals from both facilities. Besides *Clostridium* sp. MD294 and *Clostridum* sp. MD300, which are less well-colonizing GM15 animals than SPF/SOPF animals in both facilities, there was a clear general tendency of more similar fecal titers in GM15 animals as compared to their titers in SPF/SOPF animals between facilities (Fig. 7b and Supplementary Table 2) suggestive of more reproducible microbiota in GM15 animals. As a control gnotobiotic mouse model with a significantly different microbiota from GM15 (Fig. 1e), we generated Oligo-MM[12] animals in facility 2. As previously reported[27], Oligo-MM[12] gnotobiotic animals were successfully established in facility 2 and the *Bifidobacterium animalis* YL2 and *Acutalibacter muris* KB18 strains were again not detected (Supplementary Fig. 9 and Supplementary Table 3).

Then, we conducted a comparative study to evaluate the growth phenotypes of SPF, Oligo-MM[12], and GM15 mice in facility 2 on standard BD. Both GM15 and Oligo-MM[12] mice grew well compared to SPF mice (Fig. 8a, e), with Oligo-MM[12] females gaining weight slightly faster than GM15 females (Fig. 8a). Importantly, GM15 growth phenotypes (mean daily body weight and size gains) were more reproducible between facilities 1 and 2 than for SOPF/SPF animals, which significantly differed between the facilities (Fig. 8b, f).

These macroscopic observations correlated well with internal organ size: the weights of GM15 animals' brain, kidneys, liver, spleen, muscles, and adipose tissues were larger or equivalent in females (Supplementary Fig. 10a) or larger in males (Supplementary Fig. 10b) than those of SPF mice. Of note, female Oligo-MM[12] mice had less white adipose tissue (WAT) and more brown adipose tissue (BAT) than GM15 and SPF females. In GM15 males, the WAT was significantly heavier as compared to SPF and Oligo-MM[12] males. This observation was different compared to the results from facility 1, where GM15 males' visceral WAT was lighter compared to SOPF males on BD (Supplementary Fig. 8b). We observed no difference regarding the BAT among male groups, both in facility 1 and facility 2. As expected, both gnotobiotic models had a slight cecum

enlargement compared to SPF animals, yet their cecum size was significantly reduced as compared to one of GF animals (Supplementary Fig. 10a, b; and see Fig. 3g for GF). Finally, bone size was increased in both gnotobiotic models of both sexes compared to SPF animals with a more pronounced and statistically robust effect in GM15 mice (Fig. 8c, g).

Taken together these results establish that the macroscopic phenotypes of GM15 mice identified in facility 2 are similar to those from facility 1. Importantly, when comparing the results from the two facilities, we could also establish that GM15 mice growth phenotypes are more reproducible between facilities than SOPF/SPF animals. Our results also confirm that gnotobiotic models carrying a minimal microbiota compensate effectively the physiological limitations of GF mice and mimic (when not improving) the macroscopic growth phenotypes of SOPF/SPF animals. Such results correlate well with the endocrine markers of growth IGF-1 and IGFBP-3 (Fig. 8d, h) and are not due to a difference in relative food intake between conditions (Supplementary Fig. 10c). Finally, our results also suggest that each gnotobiotic model carries its own phenotypical characteristics at the steady state: a specific lower WAT-higher BAT phenotype in Oligo-MM[12] females and an increased bone growth in GM15 animals.

In facility 1, we reported that GM15 microbiota is partially protective against the deleterious effects of chronic undernutrition, as compared to SOPF mice (Fig. 6). To test the reproducibility of this phenotype in another facility, we submitted GM15 males from facility 2 to the same severe nutritional stress, by using the same nutrient DD as in facility 1 shown to trigger severe stunting after weaning (Fig. 6). This time, we also tested Oligo-MM[12] males under the same dietary regime. As in facility 1, all animals developed a macroscopic stunting phenotype and a related bone and endocrine markers alterations (Fig. 8e–h). Interestingly, as in facility 1, and despite a marked stunting phenotype, we detected an improved body weight and body size gain of GM15 males on the DD compared to SPF males (Fig. 8e, f). This observation is confirmed by an increased size of bones (Fig. 8g) and IGF-1 and IGFBP-3 levels in sera (Fig. 8h) in GM15 males as compared to SPF animals. Of note, on the DD, Oligo-MM[12] males also performed significantly better than SPF animals, a very similar phenotype to GM15 males, suggesting that this buffering effect is not a unique attribute of the GM15 model.

**The GM15 model immune and metabolic phenotypes are reproducible in another gnotobiotic facility.** To follow up on our initial phenotyping of animals from facility 1, we studied specific immune and metabolic parameters of SPF, Oligo-MM[12], and GM15 animals from facility 2. As previously reported in facility 1, the GM15 community supports IgA (both serum and fecal levels), IgG2b, IgG3, and IgE production (Supplementary Fig. 11a). Of note, we confirmed in facility 2 the similar IgE levels between GM15 and SPF animals using both Luminex and ELISA methods (Supplementary Fig. 11a, b). GM15 also supports

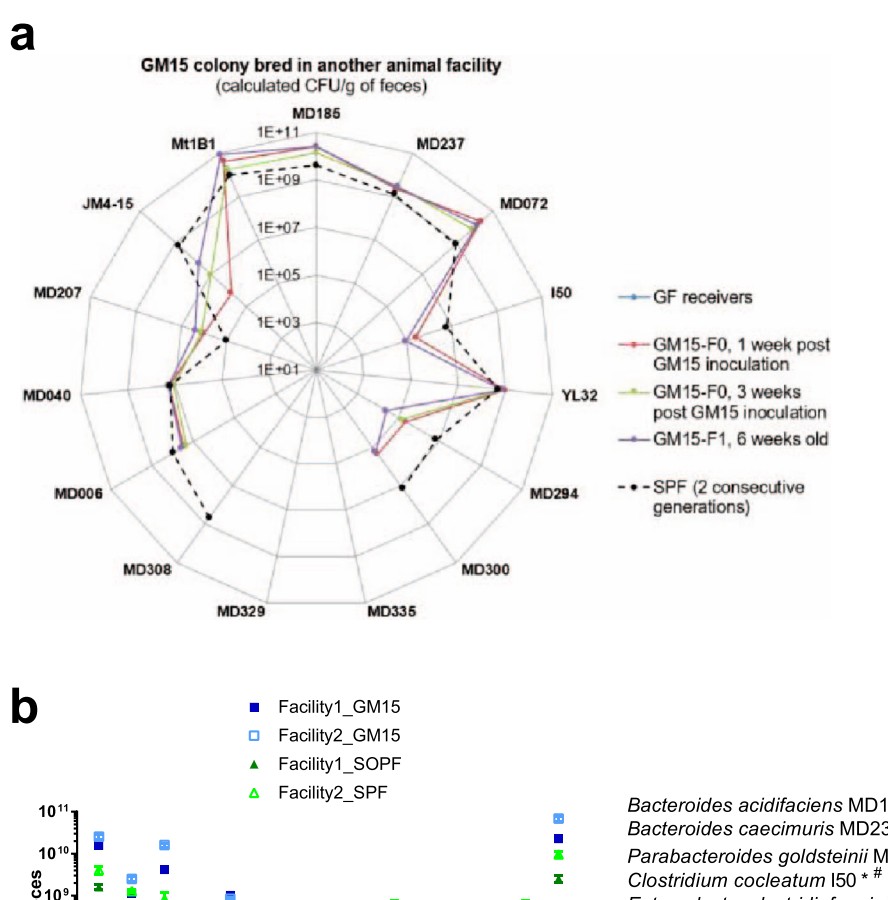

**Fig. 7 The GM15 community is reproducible in another gnotobiotic facility. a, b** The absolute quantification of each strain was determined by specific qPCR microfluidic assay. Strains MD072#, I50*#, YL32#, MD294*#, MD300*#, MD308# MD006#, MD207#, and JM4-15*# were at the detection limit of the qPCR microfluidic assay and thus were not detected in all GM15 (*) or SPF (#) samples, respectively. Strains MD335**##, MD329**##, and MD308** were below the detection limit of the qPCR microfluidic assay and thus were not detected in any GM15 (**) or SPF (##) samples, respectively. Strain YL32, obtained from the DSMZ collection, was not detected in our SOPF colony (facility 1). **a** Radar plot showing the GM15 strains distribution in feces of C57BL/ 6J GF mice colonized in facility 2 with the GM15 community (F0, $n = 9$) and bred for one generation (F1, $n = 18$). SPF group shows the distribution of each GM15 strain in the complex gut microbiota of 6–8-week-old SPF mice from two consecutive generations (F1, $n = 9$; F2, $n = 24$). **b** Grouped plot where lines and error bars represent, respectively, means and SEM of GM15 strains concentrations considering fecal samples of mice (generation F0 and F1) from facility 1 (29 GM15 and 19 SOPF) and facility 2 (27 GM15 and 33 SPF). Source data are provided as a Source Data file.

increased levels of IgM and IgG2a and supports circulating levels of IL-22 and IL-17a as in SPF animals (Supplementary Fig. 11a, c). Collectively, these results confirm, in another facility, that the GM15 community is sufficient to restore key parameters of immune system development and maturation known to be altered in GF animals. Finally, while comparing GM15 and Oligo-MM[12] models, the GM15 consortium seemed more prone to support Ig (fecal and serum IgA, serum IgG2b, IgE, IgG2a, IgG3, and IgM) and IL (IL-17a and IL-22) production. Interestingly, these features are regulated upon recognition of intestinal microbiota[45,47,48,69]. Finally, a particularity of the Oligo-MM[12] model was its lower circulating levels of IgE, IgG2a, serum IgA,

and IL-22 compared to SPF and GM15 mice, illustrating a difference in the immune phenotype of the two gnotobiotic models.

To explore the metabolic profiles of GM15, SPF, and Oligo-MM[12] animals in facility 2, we conducted [1]H NMR on plasma samples as previously performed on the plasma of facility 1 animals. We were able to quantify a total of 54 polar metabolites and 5 nonpolar metabolites (Supplementary Data 2). PCA based on quantified polar metabolites showed a sequential alignment according to sample type (GM15, SPF, or Oligo-MM[12]) on the first and second principal components (9.6% and 7.1% of total variance respectively, and 6.3% on the third component; Supplementary Fig. 11d) and calculated distance placed SPF

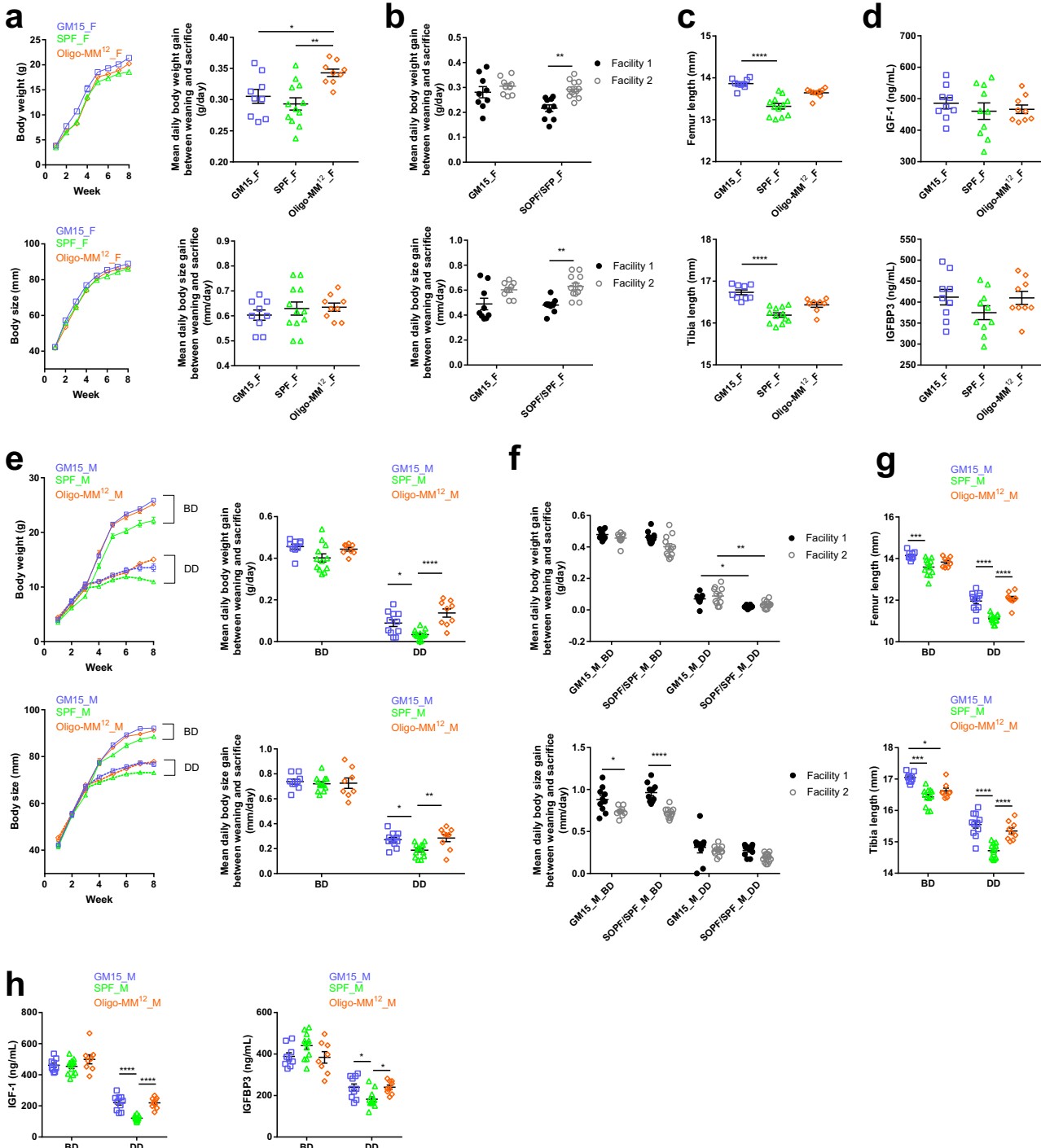

**Fig. 8 The GM15 model growth and diet-induced stunting phenotypes are reproducible in another gnotobiotic facility. a–h** Macroscopic phenotype of mice where dots, lines, and error bars represent, respectively, individual mice, means, and SEM. Female (F) and male (M), breeding diet (BD), and depleted diet (DD), respectively. **a**, **e** Body weight or size curves and mean daily gain of animals in facility 2. One-way ANOVA followed by Tukey's multiple comparison analyses (GM15 mice: 9 F, 9 M_BD, 11 M_DD; SPF mice: 12 F, 12 M_BD, 13 M_DD; Oligo-MM12 mice: 9 F, 8 M_BD, 9 M_DD). **b**, **f** Mean daily body weight and size gain comparison between GM15 and SOPF/SPF mice in facilities 1 (GM15 mice: 9 F, 10 M_BD, 9 M_DD; SOPF mice: 10 F, 10 M_BD, 10 M_DD) and 2 (GM15 mice: 9 F, 9 M_BD, 11 M_DD; SPF mice: 12 F, 12 M_BD, 13 M_DD). One-way ANOVA followed by Tukey's multiple comparison analyses. **c**, **g** One-way ANOVA analyses of femur and tibia length of animals in facility 2. One-way ANOVA followed by Dunn's and Tukey's multiple comparison analyses for F and M data sets, respectively (GM15 mice: 9 F, 9 M_BD, 11 M_DD; SPF mice: 12 F, 12 M_BD, 13 M_DD; Oligo-MM12 mice: 9 F for femur and 8 F for tibia, 8 M_BD, 9 M_DD). **d**, **h** Serum IGF-1 and IGFB3 of animals at day 56 in facility 2. One-way ANOVA followed by Tukey's multiple comparison analyses (GM15 mice: 9 F, 9 M_BD, 10 M_DD; SPF mice: 10 F, 12 M_BD, 12 M_DD; Oligo-MM12 mice: 9 F, 8 M_BD, 9 M_DD). *P < 0.05, **P < 0.01, ***P < 0.001, ****P < 0.0001. Source data are provided as a Source Data file.

sample quasi-equidistant to GM15 and Oligo-MM[12] samples, with each gnotobiotic model showing a distinct metabolic phenotype strongly resembling that of SPF animals. Polar metabolites were more prone to sex and group effects in facility 2 than in facility 1, but lipids were stable, and male to female difference (fold change > 1.3 and $p$ value < 0.05) was only seen in triacylglyceride in SPF mice from facility 2 (and in GF mice from facility 1). Like in facility 1, GM15 and SPF mice shared many similarities and a lower methanol/citrate ratio was again observed in GM15 mice. Metabolites profiles of GM15 and Oligo-MM[12] mice were also similar; nonpolar metabolites were equivalent in all mice except triacylglyceride, which was higher in SPF mice. Finally, we noticed as many differences between GM15 animals as between SOPF and SPF mice across the two facilities. Such variability might be due to the different BDs used in the two facilities, and the fact that mice were housed in isolators in facility 1 rather than in individually ventilated cages in facility 2.

Taken collectively, these results obtained in a second gnotobiotic facility confirm that gnotobiotic models carrying minimal microbial consortia such as the GM15 community mimic SPF/SOPF animal phenotypes at steady states.

## Discussion

Here we describe GM15, a simplified and controlled murine gut microbiota and its related C57BL/6J gnotobiotic mouse model. The GM15 community is composed of pure cultures of 15 strains from 7 of the 20 most prevalent bacterial families present in the fecal microbiota of C57BL/6J SOPF mice. GM15 carries significant potential for enzymatic activities in the gut and recapitulates extensively the functionalities found in C57BL/6J SOPF gut microbiome. In vivo, GM15 is stable upon adult colonization for up to 12 months, during natural transmission among nine filial generations, upon mild dietary fluctuations, can be transmitted efficiently by FMT, and can be re-established efficiently in another gnotobiotic facility. GM15 compensates breeding, growth, immune, endocrine, and metabolic limitations of GF mice and recapitulates many SOPF animals' phenotypical features in a reproducible manner in two gnotobiotic facilities.

As all experimental models, GM15 has its limitations and microbiota standardization, which is essential for establishing robust causal relationships between a microbiota configuration and a host trait, may decrease the translational potential of the observation. Despite being macroscopically similar, GM15 and SOPF animals differ in specific immune and metabolic signatures. This is not particularly surprising given the marked reduced microbial diversity that the GM15 model carries compared to SOPF microbiota (15 strains versus hundreds of species in SOPF animals). The specific immune and metabolic signatures of the GM15 model pave the way to further studies aiming at defining if the presence or absence of specific community members triggers these phenotypes. Besides, the GM15 model offers insights for gnotobiotic research as a complementary model to other murine bacteria-based models like ASF and Oligo-MM[12]. Indeed, GM15 covers more functionalities of the C57BL/6J SOPF microbiome as compared to the other two models. For instance, microbiota-mediated resistance to *Salmonella enterica* serovar Typhimurium infection may be tested in GM15 animals and compared to ASF and Oligo-MM[12] animals as recently done by Stecher and colleagues[22]. Yet, our phenotypical comparison between the GM15 and Oligo-MM[12] models reveal that both gnotobiotic animals largely mimic SPF/SOPF animal phenotypes at steady state and both show an improved response to chronic under-nutrition and as a result a milder diet-induced stunting pheno-type. Each model also carries its intrinsic phenotypical features: improved bone growth and immune maturation in GM15

animals and improved weight gain dynamics upon juvenile growth and a lower WAT/higher BAT ratio seems a hallmark of the Oligo-MM[12] model. These phenotypical differences also translate into the metabolic phenotype of these animals, which seem more distant between them, yet still close to the SPF/SOPF metabophenotype, but in their respective manner. These observations, therefore, pave the way to study the underlying symbiotic mechanisms that support these shared and/or unique phenotypes. These controlled minimal microbiota models are unique tools to study these mechanisms in mice with an advanced microbial resolution.

Taken together, our results establish that GM15 is a controlled preclinical model phenotypically similar to SOPF with the potential to ensure an increased reproducibility and robustness of preclinical studies by limiting the confounding effect of microbiota composition fluctuation and evolution. Importantly, the reduced microbial complexity of the GM15 community, the efficacy of its transfer to GF animals in different facilities, the tractability of its members, and the control offered to the scientist on its composition over time allow easy quantification and recording of short- and/or long-term gut microbiota dynamics, a current limit when using SPF/SOPF animals. Importantly, it was recently exemplified that minimal microbial communities do naturally evolve in gnotobiotic animals bred on a chow diet or upon dietary challenges[28]. In order to ensure optimal reproducibility of preclinical works, it is advisable to regularly re-establish the models using fresh frozen bacterial samples and new GF animals to avoid drift and/or selection of variants in the microbial community. The GM15 model, by its low complexity, also offers the possibility to use it as a template for establishing further complex consortia, e.g. by complementing it with representative strains of the missing prevalent bacteria family found in SOPF microbiota such as *Deferribacteraceae*, *Oscillospiraceae*, *Clostridiaceae*, and *Eubacteriaceae*. Indeed, upon manipulation of the GM15 community composition, the correlation of such dynamics with fluctuating host traits allows the establishment of causal relationships between specific microbiota members and host traits. GM15 model offers exciting perspectives for improvement. Actually, under stressful nutritional environment, the simplified GM15 microbiota performs slightly better than a complex SOPF community in terms of the physiological host response. We have previously identified a lactobacilli strain that is capable of buffering the deleterious effects of such challenge in monocolonized mice[43]. Interestingly, out of its 15 strains, GM15 microbiota contains 3 lactobacilli strains. Further genetic manipulations coupled to gnotobiotic studies focusing on modifying the lactobacilli components of GM15 may pave the way to understanding how this minimal bacterial community buffers the host response to chronic undernutrition.

In conclusion, our study establishes that the GM15 model offers possibilities for preclinical research focusing on host–microbe and microbe–microbe interactions, and how the microbiota shapes the environmental impact on health and diseases or drug efficacy.

## Methods

**Bacterial strains isolation and identification.** Fresh cecal contents and fecal pellets of C57BL/6J SOPF mice (Charles River Lab., France) were resuspended (1/10 wt/vol) in reduced broth media for direct dilution plating on agar plates and growth at 37 °C under anaerobic atmosphere (90% $N_2$, 5% $H_2$, 5% $CO_2$). *Lactobacillus johnsonii* MD006 was isolated on MRS agar. *Ligilactobacillus murinus* MD040 and *Parabacteroides goldsteinii* MD072 were isolated on Colombia CNA agar with 5% sheep blood. *Bacteroides acidifaciens* MD185 and *Irregularicoccus caecimuris* MD308 were isolated on GAM agar. *Bacteroides caecimuris* MD237 and *Limosilactobacillus reuteri* MD207 were isolated on GAM agar supplemented, respectively, with 32 μg/mL vancomycin and 32 μg/mL erythromycin. *Subtilibacillum caecimuris* MD335 and *Longibacillum caecimuris* MD329 were isolated on M2GSC agar (modified Hobson, containing (per 100 mL) 1 g of casitone, 0.25 g

of yeast extract, 0.4 g of NaHCO₃, 0.2 g of glucose, 0.2 g of cellobiose, 0.2 g of soluble starch, 30 mL of clarified rumen fluid, 0.1 g of cysteine, 0.045 g of K₂HPO₄, 0.045 g of KH₂PO₄, 0.09 g of (NH4)₂SO₄, 0.09 g of NaCl, 0.009 g of MgSO₄·7H₂O, 0.009 g of CaCl₂, 0.1 mg of resazurin, and 1.5 g of agar). *Subtilibacillum caecimuris* MD335, *Longibacillum caecimuris* MD329, and *Irregularicoccus caecimuris* MD308 were isolated from cecal contents, the rest from fecal pellets. Fecal pellets of ASF mice (Taconic, USA) were cryopreserved at −80 °C and then resuspended in reduced broth media for direct FMT in GF mice. Fresh cecal content and fecal pellets were resuspended in reduced broth media for direct dilution plating on agar plates and growth at 37 °C under an anaerobic atmosphere (90% N₂, 5% H₂, 5% CO₂). *Clostridium* sp. MD294 and *Clostridium* sp. MD300 were isolated on M2GSC agar, respectively, from cecal content and fecal pellets of ASF mice. For identification of isolates, colonies were prescreened for dereplication by MALDI-TOF MS (Vitek MS, Biomérieux) according to the manufacturer's instructions and database enrichment using RUO mode[70]. Then, genomic DNA (gDNA) was extracted from pure cultures and analyzed by 16S rRNA gene sequencing (GATC Biotech). Following Edgar's recommendation[71], a full-length 16S rRNA sequence similarity ≥99% using either NCBI blast[31], Ribosomal Database Project[72], or EzTaxon[73] programs allowed the identification of 12 isolates at the species level, and isolates MD329, MD335, and MD308, which are described as novel taxa in this study, could only be assigned to the *Lachnospiraceae* family. A more precise annotation could be given for the two isolates *Clostridium* sp. MD294 and *Clostridium* sp. MD300, respectively, as ASF356 and ASF502, since they were obtained from the defined ASF microbial consortium. *Anaerotruncus colihominis* JM4-15 (DSM-28734), *Enterocloster clostridioformis* YL32 (DSM-26114), *Clostridium cocleatum* I50 (DSM-1551), and *Escherichia coli* Mt1B1 (DSM-28618) were obtained from DSMZ.

**Description of three novel *Lachnospiraceae* strains**. The descriptions were performed using Protologger v0.99[74] and based on 16S rRNA gene sequence analysis and genome sequence analysis, including whole-proteome-based phylogenomic GBDP (Genome Blast Distance Phylogeny) analysis, percentage of conserved proteins (POCP), and differences in G + C content of DNA.

*Description of* Subtilibacillum *gen. nov.* Subtilibacillum (L. masc./fem. adj. *subtilis*, slender; N.L. neut. n. *bacillum*, rod; N.L. neut. n. *Subtilibacillum*, a slender rod-shaped bacterium). Based on 16S rRNA gene sequence similarity, the closest relative was *Kineothrix alysoides* KX356505 (90.4%). POCP analysis confirmed that strain MD335 represents a distinct genus as all POCP values to close relatives were below 50%. GTDB-Tk supported the placement of strain MD335 within a genus predicted metagenomically as "GCF_000403335.2." No antibiotic resistance genes were identified within the genome and the G + C content of genomic DNA is 43.7 mol%. The type species is *Subtilibacillum caecimuris*.

*Description of* Subtilibacillum caecimuris *sp. nov.* Subtilibacillum caecimuris (L. n. *cecum*, cecum: L. n. *muris*, mouse; N.L. gen. n. *caecimuris*, from the cecum of a mouse).

The cells are rods and strictly anaerobic. The species contains at least 297 CAZymes; however, only starch was suggested as a carbon source. KEGG analysis identified pathways for acetate production from acetyl-CoA (EC:2.3.1.8, 2.7.2.1), propionate production from propanoyl-CoA (EC:2.3.1.8, 2.7.2.1), sulfide and L-serine utilized to produce L-cysteine and acetate (EC:2.3.1.30, 2.5.1.47), L-glutamate production from ammonia via L-glutamine (EC:6.3.1.2, 1.4.1.–), and folate biosynthesis from 7,8-dihydrofolate (EC:1.5.13). This species was most commonly identified within mouse gut samples (36.9%), although subdominant at 0.35% mean relative abundance.

*Description of* Longibacillum *gen. nov.* Longibacillum (L. masc. adj. *longus*, long; N.L. neut. n. *bacillum*, rod; N.L. neut. n. *Longibacillum*, a long rod-shaped bacterium). Average nucleotide values to all close relatives were below 95%, and the best match based on 16S rRNA gene sequence similarity was *Roseburia intestinalis* AJ312385 (91.01%). POCP analysis and GTDB-Tk confirmed the creation of a novel genus, placing strain MD329 within the metagenomically predicted genus "CAG-41." No antibiotic resistance genes were identified within the genome and the G + C content of genomic DNA is 39.5 mol%. The type species is *Longibacillum caecimuris*.

*Description of* Longibacillum caecimuris *sp. nov.* Longibacillum caecimuris (L. n. *cecum*, cecum: L. n. *muris*, mouse; N.L. gen. n. *caecimuris*, from the cecum of a mouse). The cells are long rods and strictly anaerobic. Within the genome, 186 CAZymes were identified along with the predicted utilization of starch. KEGG analysis identified pathways for acetate production from acetyl-CoA (EC:2.3.1.8, 2.7.2.1), propionate production from propanoyl-CoA (EC:2.3.1.8, 2.7.2.1), and L-glutamate production from ammonia via L-glutamine (EC:6.3.1.2, 1.4.1.–). This species was most commonly identified within mouse gut samples (36.2%), although subdominant at 0.53% mean relative abundance.

*Description of* Irregularicoccus *gen. nov.* Irregularicoccus (L. masc./fem. adj. *irregularis*, irregular; N.L. masc. n. *coccus*, coccus from Gr. masc. n. *kokkos*, grain; N.L.

masc. n. *Irregularicoccus*, an irregular coccus-shaped bacterium). Strain MD308 was identified as a distinct genus to its closest relative, *Ruminococcus gnavus* X94967 (95.6%), based on 16S rRNA gene sequence similarity since GTDB-Tk was unable to match the input genome to that of a previously sequenced genome via FastANI and POCP values were below 50%. No antibiotic resistance genes were identified within the genome and the G + C content of genomic DNA is 42.77 mol%. The type species of this proposed genus is *Irregularicoccus caecimuris*.

*Description of* Irregularicoccus caecimuris *sp. nov.* Irregularicoccus caecimuris (L. n. *cecum*, cecum: L. n. *muris*, mouse; N.L. gen. n. *caecimuris*, from the cecum of a mouse). The cells are rods that separate into coccoid forms, and are strictly anaerobic. The number of CAZymes identified within the genome was 226, facilitating the predicted utilization of cellulose and starch as carbon sources. KEGG analysis identified pathways for acetate production from acetyl-CoA (EC:2.3.1.8, 2.7.2.1), propionate production from propanoyl-CoA (EC:2.3.1.8, 2.7.2.1), sulfide and L-serine utilized to produce L-cysteine and acetate (EC:2.3.1.30, 2.5.1.47), L-glutamate production from ammonia via L-glutamine (EC:6.3.1.2, 1.4.1.–), and folate biosynthesis from 7,8-dihydrofolate (EC:1.5.13). This species was most commonly identified within mouse gut samples (68%), although subdominant at 0.69% mean relative abundance.

**Culture conditions**. Freshly grown cultures of individual bacterial strains were supplemented with 20% glycerol prior to cryopreservation at −80 °C. Each culture was systematically validated for culture purity and identity by MALDI-TOF. Culture media and material were introduced in the anaerobic chamber at least 2 days before use. Anaerobic bacterial strains were grown in GAM, except *Clostridium* sp. MD300, *Irregularicoccus caecimuris* MD308, *Subtilibacillum caecimuris* MD335 and *Longibacillum caecimuris* MD329 in M2GSC, and *Anaerotruncus colihominis* JM4-15 in Bifidobacterium medium. For mouse colonization and absolute quantification of bacteria, a single colony of each of the 15 bacterial strains was grown and amplified at 37 °C. Each bacterial pellet was resuspended in the medium, 1 mL was cryopreserved with 10% glycerol, 1 mL was centrifuged, and the bacterial pellet was stored at −20 °C for gDNA extraction, and the rest was used for enumeration by dilution plating on agar plates. A frozen mixture of the GM15 bacterial community containing the 15 individual strains at an equivalent concentration (6.67E + 06 CFU (colony-forming unit)) was prepared to enable easy inoculation.

**Whole-genome sequencing**. DNA samples from the 15 bacterial cultures were prepared for WGS, using the Nextera XT DNA library preparation kit (Illumina, California, USA) according to the manufacturer's recommendations. The resulting libraries were checked for their quality using the High-sensitivity DNA chip using the Agilent 2100 Bioanalyzer (Waldbroon, Germany) and quantified using the QuantiFluor One dsDNA kit (Promega). Paired-end (2 × 300 bp) sequencing was performed on a MiSeq sequencer using the MiSeq v3 kit (600 cycles) (Illumina, California, USA).

**De novo genome assembly**. After a quality control with FastQC v0.11.7[75], the paired-end reads were assembled de novo using the "A5-miseq" assembly pipeline v20160825[76], comprising the following steps: adapter trimming, quality trimming and filtering, error correction, contiging, and scaffolding. The 15 de novo assemblies resulted in draft genomes composed of a few scaffolds (from 30 to 268) with high N50 values (from 13,099 to 943,892). Genomes were then ordered using Mauve v2.4.0[77] and annotated with PGAP of the NCBI database. Default parameters were used for all software tools.

**Taxonomic annotation**. WGS generated data have been trimmed and quality controlled by AfterQC software v0.9.1[78]. A k-mer counting strategy with the Centrifuge software v1.0.3[29] has been privileged to infer taxonomy, and reads were confronted to the RefSeq complete genome database Release 84[30], with bacteria, archaea, and viruses domains and the mouse representative genome (taxid 10090), in order to estimate the amount of host DNA contamination and remove it from sequenced data.

**Genomic functional analysis**. Genes were predicted and translated into protein sequences using Prodigal v2.6.2[79]. Marker genes were searched using the HMM3 package v3.0[80]. Predicted protein sequences of genomes were blasted against the KEGG microbial database Release 2018-01-29[81], which contains 13 million proteins sequences and trimmed with the following parameters: best hit with an expected value threshold <10⁻⁵. The matrix obtained was consolidated into KEGG orthologs counts (KO, which represent sets of genes that have sequence similarities and exert the same function), into KEGG modules (which represent short enzymatic pathways, involving few proteins and doing a targeted function), and into KEGG pathways (i.e., large metabolic pathways). KO were analyzed for their presence or absence among genomes. The modules were analyzed for their completion (four levels: full, lack 1 enzymes, lack two enzymes, absent), and only modules with a score of 3 or 4 were presented and integrated for their KO counts. The KEGG pathways were analyzed for their number of related KO counts

affiliated to them. A list of functions of interest has been designed and their presence among genomes has been analyzed in detail (Supplementary Data 1). Because pathways and functions of interest did not have the same number of KO of interest and a different distribution among the genomes, functional data were normalized among each function in order to obtain values that can be comparable. For each function/pathway, the number of different KO was normalized by the total number of KO retrieved. Data were then log-transformed +1. Clusterization of both functions and communities was performed using Euclidian distance and ward's method, and a k-mean clustering was performed in order to define the community clusters.

**Identification of specific regions for primers design**. NUCmer, a part of the MUMmer package v4.0[82], was used to perform pairwise alignment of the 15 genomes. Based on these alignments, PanCake v9.1[83] was used with default parameters to identify specific regions of each genome. Specific regions with a length of 200 bp were extracted, meeting the following criteria: GC content between 48 and 52% distance to a border of a scaffold higher than 300 bp, unique in the draft genome. The specificity of each 200 bp region was double-checked with BLAST v2.7.1[31] on the 15-genome database and on the NCBI *nr* database accessed in February 2018. The design of primers on the specific regions was performed by Fluidigm. The primer specificity was checked with BLAST.

**Animal experiment**. *Facility 1*: Mice were bred according to standardized procedures in the gnotobiology unit of BIOASTER at the ANSES animal facility (Lyon, France), housed in sterilized positive pressure isolators (Noroit) under a 12 h light/dark cycle at 22 ± 2 °C and 50 ± 30% of humidity, and fed ad libitum with irradiated R03-40 diet (3395 kcal/kg, 25.2% kcal proteins, 61.3% kcal carbohydrates, 13.5% kcal lipids; Safe), and autoclaved water. Irradiated corn-cob granules (Safe) were used as bedding. Sterile enrichment was provided in all cages and was constituted by cotton rolls as nesting material, poplar bricks, and a polycarbonate red mouse igloo (Safe). Nesting material and poplar bricks were renewed every 2 weeks. All breeders were mated by the trio (two females and one male) between 8 weeks and 6 months of age, and all mice were weaned at 4 weeks after birth. C57BL/6J SOPF mice were obtained from Charles River Lab. The fecal samples used for the in silico design of the initial metagenome were collected in-house from four 2-month-old littermates housed in different cages from weaning at 3 weeks old. C57BL/6J GF mice were produced in-house by an aseptic hysterectomy of a C57BL/6 J SOPF female, and neonates were fostered on C3H GF mothers (CDTA). Axenic status was assessed weekly by gram staining and cultures of fecal suspension on solid and liquid media. GM15 founders were 8-week-old C57BL/6 J GF mice colonized by oral gavage with 215 µL of the fresh frozen GM15 bacterial community, twice at a 48-h interval. GM15 microbiota composition was analyzed by qPCR microfluidic assay from feces collected at 6-week-old. Alternative diet R04-40 (3339 kcal/kg, 19.3% kcal proteins, 72.4% kcal carbohydrates, 8.4% kcal lipids; Safe) was given at 8-week-old GM15 mice for 4 weeks. FMT was done by oral gavage with a suspension of fresh fecal pellets administered to 7-week-old C57BL/6J GF mice twice at a 48-h interval. For undernutrition experiments, GM15 and SOPF male mice were bred and randomly assigned at day 21 after birth to be given either the above R03-40 diet or a custom-made low-protein diet (3500 kcal/kg, 4.7% kcal proteins, 90.1% kcal carbohydrates, 5.3% kcal lipids, Envigo) for 5 weeks after weaning. Mice were killed by cervical dislocation and biocollections were performed aseptically. All animal procedures were approved by the French MESR and the ANSES/ENVA/UPEC ethics committee (APAFIS#4529-2016022616404045v3; APAFIS#785-2015042819315178v2; APAFIS#18918-2019020118003843v3) and were conducted in accordance with the National and European legislation on protection of animals used for scientific purposes.

*Facility 2*: BALB/c SPF mice (Supplementary Fig. 7), bred in the Laboratory of Gnotobiology for more than 10 generations, were kept in individually ventilated cages (Tecniplast, Italy), exposed to 12 h light/dark cycles, and fed with 25 kGy irradiated BD (Altromin 1414, Altromin, Germany) ad libitum. Animal procedures were approved by the committee for the protection and use of experimental animals of the Institute of Microbiology of the Czech Academy of Science (approval ID: 117/2013). Ten- to 11-week-old BALB/c SPF mice were mated. After the delivery, the litter size was reduced to six offspring per dam. On day 21 after birth, male and female mice were weaned either on the BD (Altromin 1414 containing 25.1% of crude protein and 9% of fat, metabolizable energy 3646 kcal/kg) or the nutritionally DD low in proteins (8.6%), fats (2.4%), and vitamins (modified from Altromin C1003, metabolizable energy 3580 kcal/kg)[43]. On day 56 after birth, mice were weighted and body length was measured. C57BL/6J GF mice, bred in the Laboratory of Gnotobiology for more than ten generations, were kept under sterile conditions in positive pressure Trexler-type plastic isolators on sterile Abedd Espe LTE E-002 bedding (Abedd, Germany), exposed to 12 h light/dark cycles at 22 ± 2 °C temperature and 40–60% humidity, and supplied with autoclaved tap water and 50 kGy irradiated sterile pellets (BD: SSNIFF mouse breeding fortified diet V1124-300, 3338 kcal/kg, 27% kcal proteins, 61% kcal carbohydrates, 12% kcal lipids) ad libitum. Axenicity was assessed every 2 weeks by confirming the absence of bacteria, molds, and yeast by aerobic and anaerobic cultivation of mouse feces and swabs from the isolators in VL (Viande-Levure), Sabouraud-dextrose, and meat-peptone broth and subsequent plating on blood, Sabouraud, and VL agar plates. C57BL/6J SPF mice were kept individually

ventilated cages (Tecniplast, Italy), exposed to 12 h light/dark cycles, and fed with the same sterile diet as GF counterparts. Animal procedures were approved by the committee for the protection and use of experimental animals of the Institute of Microbiology of the Czech Academy of Science (approval ID: 3/2019). GM15 founders were 8-week-old C57BL/6J GF mice orally gavaged with 150 µL of the frozen GM15 bacterial community twice at a 48-h interval. GM15 microbiota composition was analyzed by qPCR microfluidic assay as described for facility 1. Oligo-MM[12] gnotobiotic mouse line was established in 2018[27]. Colonized mice were transferred to an experimental isolator and their offspring were used for the experiments. GM15, Oligo-MM[12], and SPF mice were bred and randomly assigned at day 21 after birth to be given either the SSNIFF V1124-300 diet (males or females) or a custom-made low-protein diet (3500 kcal/kg, 4.7% kcal proteins, 90.1% kcal carbohydrates, 5.3% kcal lipids, Envigo) (males only) for 5 weeks after weaning. For the measurement of body length, mice were briefly anesthetized by isoflurane (Piramal Healthcare, UK). Weight was measured thrice and the scales (Acculab mini PP201, Sartorius, Germany) were tared between the measurements. Males and females on the BD were sacrificed between 9 and 11 a.m. without food removal. Males fed the DD were sacrificed between 1 and 3 p.m. after 5 h starvation. Mice were killed by cervical dislocation and biocollections were performed aseptically.

**gDNA extraction from cecal contents and fecal pellets**. *GM15, SOPF/SPF mice*: Cecal and fecal gDNA were extracted using the DNeasy® PowerLyzer® PowerSoil® Kit (Qiagen) following the manufacturer's instructions with modifications. Samples (approximately 0.1 g) were heat-treated at 65 °C for 10 min and 95 °C for 10 min, before a double bead beating at 30 Hz for 5 min. Fifty microliters of DNA were obtained with two consecutive elutions.

*Oligo-MM[12] mice*: Fecal samples were collected and frozen at −80 °C. Fecal gDNA was extracted using the Nucleospin DNA stool kit (Macherey-Nagel) according to the manufacturer's instructions with the following modification: The initial bead-beating step was performed on TissueLyzer (Qiagen) at 40 Hz for 7 min:30 s. gDNA was eluted with 60 µL of SE buffer.

**Quantitative PCR microfluidic assay**. In order to quantify specific and global bacteria load per g of cecal or fecal samples (wet weight), qPCR microfluidic assay was performed using respectively specific primers for GM15 and "universal" primers amplifying the genes encoding 16S rRNA from most bacteria (Supplementary Table 1). We used the "universal" primers UniF/R targeting highly conserved sequences within the *E. coli* 16S rRNA gene (1542 nucleotides) and known as a generic primer set to amplify the genes encoding 16S rRNA from all bacteria[84]. UniF is complementary to nucleotides 1047 through 1067, while UniR is complementary to nucleotides 1174 through 1194 (Supplementary Table 1). Amplicons generated using these primers range between 60 and 99 bp. qPCR microfluidic assays were conducted in 48.48 Dynamic Array™ IFCs (integrated fluidic circuits) for EvaGreen Fast Gene Expression on a Biomark HD (Fluidigm) according to the manufacturer's instructions, with complementary DNA diluted 100-fold, preamplified with pooled primers, and diluted again 100-fold. Each IFC included triplicate reactions per DNA sample, standards, and negative control. Standards were generated by serial dilution of a gDNA extract from pure bacterial cultures of known concentration. The efficiency of each qPCR reaction was calculated based on the slope of standard curves and within the range of 78–107%. An equivolume mixture of standards was used to normalize data between runs. Due to the different individual detection limits of the qPCR assay for each primer, the detection limit for GM15 ranged between $2.73 \times 10^2$ and $6.57 \times 10^5$ CFU/g (Supplementary Table 1).

**qPCR of bacterial 16S rRNA Genes**. qPCR was performed as follows: Oligo-MM[12] strain-specific 16S rRNA primers (Supplementary Table 3) and hydrolysis probes were synthesized by IDT (Integrated DNA Technologies, USA). Standard curves using linearized plasmids containing the 16S rRNA gene sequence of the individual Oligo-MM[12] strains were used for absolute quantification of 16S rRNA gene copy numbers of individual strains. 16S rRNA gene copy numbers were normalized to equal volumes of extracted DNA, assuming that DNA extraction is equally efficient between different samples. PCR reaction of 20 µL contained 0.2 µL of 30 µM working solution of each primer; 0.2 µL of 25 µM working solution of the corresponding probe; 6.9 µL H$_2$O; 10 µL of 2x PrimeTime® Gene Expression Master Mix (IDT) and 2.5 µL of gDNA of concentration 2 ng/µL. PCR conditions were 95 °C for 3 min, followed by 45 cycles of 95 °C for 15 s and 60 °C for 1 min with fluorescent measurement. PCR reactions were run on qTOWER³ G touch (Analytik Jena) and analyzed using qPCRsoft software v4.1 (Analytik Jena). Standards and samples were assayed in duplicate and monoplicate, respectively.

**Sample preparation for immunophenotyping**. For flow cytometry analyses, whole blood was collected on an EDTA tube. Spleen, thymus, MLNs, and PPs were collected in RPMI (Gibco). Single-cell suspensions were achieved using a 100 µm cell strainer (Becton Dickinson) and a 5 mL syringe plunger in RPMI supplemented with 2% heat-inactivated fetal bovine serum (Sigma) and 100 µg/mL DNASE1 (Roche). Cells were then spun at 400 × *g* for 5 min at room temperature. The medium was discarded and cells were washed using 5 mL of supplemented

medium. For whole blood, spleen, and thymus samples, red blood cells were lysed by resuspension in 1 mL PharmLyse 1X (Becton Dickinson) for 10 min. Cells were then spun at $400 \times g$ for 5 min at room temperature. The lysing solution was discarded and cells were washed using 2 mL of PBS (Gibco). Cells were pelleted a second time and resuspended in PBS supplemented with 2% heat-inactivated fetal bovine serum (Sigma). Numeration and viability were determined using propidium iodide marker exclusion and MACSQUANT flow cytometer (Miltenyi). Cells were then resuspended to a working concentration of $10^6$ cells/tube for organs and 100 μL/tube for whole blood, and analyzed by flow cytometry.

Whole blood was collected on a dry Eppendorf tube for sera analysis. Sera were obtained by centrifugation $2000 \times g$ for 15 min at 4 °C and stored at −20 °C before Luminex and ELISA analyses. Feces were collected in Eppendorf low-binding tubes and stored at −80 °C before Luminex analysis. Feces supernatant was obtained by disrupting 100 mg feces in 1 mL PBS-Protease Inhibitor 1X (Sigma) using Lysing Matrix E Tube (MP Biomedicals) and Fast Prep homogenizer (MP Biomedicals). Samples were spun at $8000 \times g$ for 15 min at 4 °C and supernatants were collected for IgA Luminex analysis.

**Flow cytometry.** A total of $10^6$ cells or totality of cells for some PP samples were stained for surface markers. Leukocytes were stained using anti-CD3 BV421 (clone 145-2C11, #562600, 1:40) from Becton Dickinson, and anti-CD45 Viogreen (clone 30F11, #130-102-412, 1:40), anti-CD4 PE (clone REA604, #130-109-414, 1:20), anti-CD8a PE Vio615 (clone REA601, #130-109-251, 1:10), anti-CD45R/B220 PE Vio770 (clone RA3-6B2, #130-102-308, 1:20), anti-CD335 APC (NKp46) (clone 29A1.4.9, #130-102-347, 1:10), and anti-CD11b APC Vio770 (clone REA592, #130-109-288, 1:20) or anti-CD11c APC Vio770 (clone REA754, #130-110-704, 1:100) from Miltenyi Biotec Gmbh. Viable cells were selected using Zombie Green Fixable Viability kit (BioLegend #423111, 1:100). T cells were identified as CD45+, CD3+, and CD4+ or CD8+ cells; B cells were identified as CD45+, CD3−, and CD45R/B220+ cells. NK cells were identified as CD45+, CD3−, CD45R/B220−, and NKp46+ cells. Monocytes were identified as CD45+, CD3−, CD45R/B220−, and CD11b+ cells in the spleen and whole blood samples. DCs were identified as CD45+, CD3−, CD45R/B220−, and CD11c+ cells in the spleen, PP, and MLN samples. Cells were analyzed using a MACSQuant Ten Flow cytometer with MACSQuantify software v2.8.1618.16380 (Miltenyi) and raw data were analyzed using FlowJo software v10.4.2 (Tree Star, Becton Dickinson). CD45+ cell counts were normalized by performing the analysis on 1 M of viable cells that were stained. For CD45+ cell count comparison in PPs, CD45+ cell counts were performed on 1 M of viable cells that were stained when possible and on total viable cells stained when <1 M cells were isolated; thus, no statistical analysis has been performed on these data. Data normalization was not possible for the whole blood sample, thus CD45+ cell count comparison has not been performed for this compartment. For frequency results, data are represented as a percentage of CD45+ cells for all organs. Samples were assayed in monoplicate.

**Luminex metabolic, Ig, and cytokine panels analysis.** Serum concentrations of Metabolic Panel (ghrelin, glucagon, insulin, and leptin) were determined using the Mouse Metabolic Magnetic Bead Panel Milliplex MAP kit (Millipore). Samples were not diluted and the assay was performed according to the manufacturer's instructions. Serum concentrations of Ig Panel (IgA, IgG1, IgG2a, IgG2b; IgG3, IgM, IgE) and IgA in feces supernatant was determined using the Mouse Ig Isotyping Magnetic Bead Panel Milliplex MAP kit (Millipore) and Mouse IgE Single Plex Magnetic Bead Milliplex MAP kit (Millipore). Samples were diluted 1:12,500 (IgA, IgG1, IgG2a, IgG2b; IgG3, IgM), 1:100 (IgE), and 1:100 (IgA in feces supernatant). Assays were performed according to the manufacturer's instructions. Serum concentrations of Cytokine Panel (IL-1b, IL-10, IL-12p70, IL-17a, IL-17f, and IL-22) were determined using the Mouse Th17 Magnetic Bead Panel Milliplex MAP kit (Millipore). Samples were not diluted and the assay was performed according to the manufacturer's instructions. Samples were assayed in monoplicate. The known intratechnical error (%CV) per assay was 7% for insulin, 8% for glucagon, 6% for leptin, 15% for ghrelin, 10% for IgA, 6% for IgG1, 8% for IgG2a, 7% for IgG2b, 5% for IgG3, 5% for IgM, 2% for IL-1b, 3% for IL-10, 2% for IL-12p70, 3% for IL-17a, 3% for IL-17f, and 3% for IL-22.

**Corticosterone, IGF-1, IGFBP-3, and total IgE ELISAs.** Serum concentrations of corticosterone were determined using the Corticosterone ELISA kit (Abnova). Samples were diluted at 1:50 and the assay was performed according to the manufacturer's instructions. Serum concentrations of IGF-1 and IGFBP-3 were determined using the Mouse/Rat IGF-1 Quantikine ELISA kit (R&D Systems) and IGFBP-3 DuoSet ELISA kit (R&D Systems), respectively. Samples were diluted at 1:500 and the assay was performed according to the manufacturer's instructions. Serum concentrations of total IgE were quantified with BD OptEIA™ Mouse IgE ELISA Set (BD Biosciences, San Chose, Canada) with samples diluted 1:100 according to the manufacturer's instructions. Samples were assayed in monoplicate.

**Sample preparation for metabophenotyping**

*Polar metabolite preparation.* It is known that the quantification of some plasma metabolites, such as tryptophan and tyrosine, is biased since they bind to albumin, a highly abundant protein in plasma[85]. In addition, protein precipitation methods

with organic solvents can induce loss of volatile metabolites and overlay of residual broad resonances of lipids with some polar metabolites[86]. Thus, the polar and nonpolar metabolites extraction from the same plasma sample was adapted according to previously described sequential approaches[87,88]. First, we deproteinized the plasma samples by acidified ultrafiltration in order to increase desorption yields of aromatic amino acids and then, quantify the polar metabolites. The concentration of the formic acid was optimized to allow desorption of the metabolites like tryptophan from plasma albumin. In practice, frozen mice plasma samples were thawed in thermoshaker Eppendorf (10 min, 10 °C, 1000 r.p.m.). The entire amount of plasma (about 180 μL) was filtered using a 0.2 μm centrifugal tube from VWR (5 min, 10 °C, $10,000 \times g$). Next, the lipoprotein removal was performed by mixing 150 μL of filtered plasma with 50 μL of milli-Q water and 10 μL of deuterated formic acid (0.1% final concentration) in a clean 10 kDa cut-off ultracentrifugation tube (VWR) using thermoshaker Eppendorf (10 min, 10 °C, 750 r.p.m.), and then by centrifugation (30 min, 10 °C, $10,000 \times g$, soft ramp). Deuterated formic acid was used instead of protonated form to decrease the exogenous NMR signal and allow the quantification of the endogenous formate. Clean 10 kDa cut-off ultracentrifugation tubes were prepared by recovering filters from tubes, transferring them into a 250 mL Duran bottle plunged in milli-Q water, and then sonicated in a bath for 10 min, repeated five times. After the last step, the 10 kDa cut-off ultracentrifugation filters could be stored in milli-Q water for at least 3 months. Just before use, a nitrogen stream was used to discard any traces of milli-Q water. We observed that this ultrasonication procedure removes better the residual glycerol than the supplier's protocol and gives higher yields of ultrafiltered plasma. Indeed, glycerol was completely discarded in the ultrasonicated tubes ($N = 5$; average = under detection limit), while it was still present after centrifugation following washes with either sodium hydroxide 0.05 N followed by four washes with milli-Q water ($N = 5$; average = 0.02 mM; standard deviation = 0.02 mM) or ethanol 70% followed by four washes with milli-Q water ($N = 5$; average = 0.05 mM; standard deviation = 0.01 mM), or five control washes with milli-Q water ($N = 5$; average = 0.02 mM; standard deviation = 0.01 mM). In addition, we observed that the use of ethanol contributed to contamination since traces of ethanol remain. Finally, regarding the mechanical properties of the filters, we also observed that after five washes and centrifugation at $10,000 \times g$, filters collapsed during plasma filtration.

Vortexed 10-kDa-filtered plasma samples (about 135 μL) were transferred in a 0.5 mL 96-well plate (Agilent) and mixed on thermoshaker Eppendorf (1.5 min, 10 °C, 650 r.p.m.) with 45 μL of internal standard solution DSS-d6 (1.54 mM), which also contains the pH-reference standard DFTMP (4 mM). The internal standard solution DSS-d6 was prepared with phosphate buffer (0.6 M, 60:40 v/v $H_2O$:$D_2O$, pH = 7.4). Finally, 155 μL of the resulting sample solutions were transferred in 3 mm SampleJet NMR tubes. Since the DSS-d6 solution might be unstable during long time storage (hydroscopic properties, trimethylsilyl fragment degradation, etc.), the DSS-d6 concentration was calibrated by $^1$H NMR using sodium succinate dibasic hexahydrate standard solution, a compound with better stability properties, in order to guarantee data accuracy. This protocol was systematically applied including for blank samples, which were prepared by replacing the plasma with Milli-Q water, and for quality controls ($n = 7$), which were prepared using commercially available mouse plasma.

*Nonpolar metabolite extraction after ultrafiltration.* Then, the lipoproteins on the 10 kDa filters were further diluted with 150 μL phosphate-buffered solution (1 M, pH = 7.4), mixed on thermoshaker Eppendorf (10 min, 10 °C, 750 r.p.m.), transferred in clean Eppendorf tubes, and extracted with 400 μL methanol-dichlormethan (1:2 v/v). Samples were centrifuged (5 min, 10 °C, $10,000 \times g$) for better phase separation. The dichlormethan layer was transferred in clean Eppendorf tubes and the aqueous phase was extracted again with dichlormethan. The pooled organic phase was evaporated under a nitrogen stream. The dry lipidic residue was dissolved with 200 μL deuterated chloroform containing 0.03% TMS internal standard and 155 μL of the resulting solution was transferred in a 3-mm SampleJet NMR tube. The TMS concentration was calibrated by NMR using 1,3,5-tri-tert-butylbenzene standard solution and it was found to be 0.435 mM.

*Metabolite analysis and quantification.* The 1D $^1$H NMR spectra were acquired at 298 K with a 600 MHz Ascend (Avance III HD) spectrometer from Bruker equipped with a 5 mm QCI cryoprobe. Samples were assayed in monoplicate. All samples were stored at 6 °C in the SampleJet autosampler. Polar metabolites were analyzed using *noesygppr1d* pulse sequence. For each spectrum, 128 scans were collected into 32k data points within 14 p.p.m. spectroscopic width and a recycling delay of 4 s. The mixing time was calibrated to 50 ms and the acquisition time was 3.9 s. The nonpolar metabolites' NMR spectra were acquired using *zg30* pulse sequence. The spectra were recorded using 256 scans, into 32k data point, and a spectroscopic width of 20 p.p.m. The relaxation delay was 4 s. The FIDs were zero-filled to 64k points and Fourier-transformed using a 0.3 Hz exponential line-broadening function. All spectra were aligned to DSS-d6 and TMS, respectively, internal standard. The concentrations of the polar and nonpolar metabolites were quantified using Chenomx NMR suite 8.6. The Chenomx software was also applied for spectra binning of $10^{-3}$ p.p.m. width for each bin. The triacylglycerol, phosphatidylcholine, lysophosphatidylcholine, sphingomyelin, free cholesterol, and cholesterol ester quantification was carried out using an in-house lipid database within Chenomx Compound Builder module based on authentic lipid standards

NMR spectra. The lipids database NMR spectra were recorded using the same parameters as described above for nonpolar metabolites.

*Metabolomics data analysis.* NMR data preprocessing included normalization of analytical batch effects using internal standards, PCA analysis to assess the relative contribution of each effect of the data by measuring the Mahalanobis distance between each group and the associated significance using a Fisher test, confirmed by supplementary multivariate supervised analyses (partial least squares-discriminant analysis (PLS-DA)) performed using MATLAB v2019b and v2021a. Value under the limit of detection, when the signal to noise was below 3, was replaced by 0 at the NMR data quantification step with Chenomx NMR suite 8.6. Thus, no values were excluded during the statistical analysis, and there were no missing values treatment and transformations applied. Then, discriminant analyses were performed using the PLS-DA algorithm to extract metabolomics signatures[89,90]. A variable selection algorithm based on Elastic-Net (MATLAB) was used to improve model performance by selecting the most significant metabolic signatures that explain the groups (GF, GM15, and SOPF). The statistical performances of the regression models were assessed using the balanced error rate with and without cross-validation (E2 and CV-E2) and permutation tests (MATLAB). Permutation tests consisted in building the null distribution of the balanced error rate E2 by randomly permuting observations. Regression models were thus challenged by testing if the cross-validation error rate CV-E2 is significantly different from the null distribution with a $p$ value < 0.05. Metabolites involved in the cross-validated signatures were ranked by order of importance in the PLS-DA model using their VIP (variable importance in projection) scores. The biplot allows projecting onto the two first components, the samples and the metabolites that significantly discriminate each sample group. Metabolites that are positively correlated (or positively contribute) to a sample group will point to the direction of this group. They will point to the reverse direction for a negative correlation.

*Statistics.* Reproductive performance and body growth were analyzed, respectively, by one- and two-way ANOVA. Phenotyping data impacted by age, filial generation, or sex were analyzed by the $F$ test for multiple linear regressions (R v3.4.2), otherwise by one-way ANOVA followed by Tukey's multiple comparison parametric test or Kruskal–Wallis and Dunn's multiple comparison nonparametric test after D'Agostino et Pearson test for data set normality assessment (GraphPad Prism v7.05) or two-way ANOVA adjusted for Sidak's post hoc test for multiple comparisons (GraphPad Prism v8.20). Heatmaps of Fig. 1d, e and correlation matrix in Supplementary Fig. 8h were performed using GraphPad Prism v7.05. Box plots and heatmap of Supplementary Fig. 1 were performed using R v3.6.1. PCA and PLS-DA of Fig. 5a, b and Supplementary Fig. 11d were performed using MATLAB v2019b and v2021a. PCA of Supplementary Fig. 6 was performed using SIMCA v13.0.3. PCA of Fig. 6g was performed using R v3.4.2 and the ade4 package v1.7-18[91].

## Data availability

*Anaerotruncus colihominis* JM4-15 (DSM-28734), *Enterocloster clostridioformis* YL32 (DSM-26114), *Clostridium cocleatum* I50 (DSM-1551), and *Escherichia coli* Mt1B1 (DSM-28618) were obtained from DSMZ. All other bacterial strains isolated by BIOASTER, and GM15 mice, are assets controlled by BIOASTER and therefore distribution is managed by BIOASTER. Any request shall be sent to Marion Darnaud at the following email address: gnotobiology@bioaster.org. All demands will be examined and transfer of bacterial strains and/or GM15 mice will be possible under specific MTA conditions. The 15 assembled genomes and the corresponding sequencing reads have been deposited in the DDBJ/ENA/GenBank data banks and the Sequence Read Archive, respectively, under the BioProject number PRJNA551571[32]. The list of KEGG modules and clusters referenced in this study, and obtained using KEGG microbial database Release 2018-01-29, are available in Supplementary Data 1. NMR raw data are accessible at the NIH Common Fund's National Metabolomics Data Repository website, the Metabolomics Workbench, https://www.metabolomicsworkbench.org (supported by NIH grant U2C-DK119886), where it has been assigned Project ID PR001164. The data can be accessed directly via its project DOI: 10.21228/M87T3G. Source data are provided with this paper.

## Code availability

The code necessary to reproduce our metabolomics analysis is available at https://github.com/bioaster/biotracs-m-atlas.

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

## Acknowledgements

We thank Julie Henry, Leanne Goncalves, Christelle Boisse, and Gustavo Stadthagen Gomez for help with bacterial cultures and gram staining, Cécile Chauvel, May Taha, Joséphine Abi Ghanem, and Adrien Villain for help with biostatistics, and Vincent Thomas, Gianfranco Grompone, Laurent Beloeil, and Alain Troesch for valuable comments and discussions, and Jaroslava Valterová and Šárka Maisnerová for technical assistance. This work was supported by the French Government as part of the Programme des Investissements d'Avenir (grant no. ANR-10-AIRT-03). Research in F.L. lab was supported by ENS de Lyon, CNRS, the FINOVI foundation, and the Fondation pour la Recherche Médicale (Equipe FRM DEQ20180339196). M.S. acknowledges the support of the Czech Science Foundation (grant no. 18-07015Y) and EMBO Installation grant.

## Author contributions

M.D., F.L. and A.T. conceived the project, analyzed, interpreted, and integrated all the data. M.D. and F.L. wrote the manuscript. M.D. prepared figures. L.B. and J.Y. achieved in silico design. M.D. performed bacteria isolation and identification, gDNA extraction, developed the qPCR microfluidic assay, and analyzed macroscopic phenotyping data. A.S. carried out WGS, C.E. identified specific regions for primer design, and A.M. performed taxonomic annotation and functional bioinformatic analyses. P.B., H.D., M.D., T.N., D.S. and M.S. performed animal experiments. B.S. provided the Oligo-MM[12] consortium. T.N. performed the Oligo-MM[12]-related gDNA extraction and consortium analysis with the help of B.S. C.C., A.D., S.P. and M.S. performed the immunophenotyping, analyzed, and interpreted data. A.B. and D.A.O. performed the metabophenotyping, analyzed, and interpreted data. M.S. helped design the diet-induced stunting protocol. F.D.V. performed, analyzed, and contributed to the text and figures related to the stunting model experiments. A.-L.B. provided technical support during the stunting model experiments. All authors read and approved the final manuscript.

## Competing interests

The authors declare no competing interests.
