## [Peer Review File · Nature Communications]

REVIEWER COMMENTS

Reviewer #1 (Remarks to the Author):

The manuscript “A standardized gnotobiotic mouse model harboring a minimal 15-member mouse gut microbiota recapitulates SOPF phenotype” by Darnaud et al. describes the generation of the minimal microbiome GM15, which consists of 15 bacterial species, in an attempt to standardize microbiome-related experimental research. Mice colonized with GM15 phenotypically resemble SPF mice in some aspects. The GM15 mice were also less sensitive to protein malnutrition. The authors conclude that (i) the GM15 model is a more stable experimental model than SPF, which should improve reproducibility in microbiome experimentation, and that (ii) the GM15 model outperforms other minimal microbiomes, e.g. ASF or OMM. The topic of this study is of considerable interest. However, there are also significant concerns/limitations that should be addressed to substantiate these main findings:

1.) Four mice were used to generate the initial metagenome. How representative were those four animals (2 males, 2 females)? Were these littermates, from the same or different cages? How old?

2.) In the KEGG Orthology analysis, how much of the genome was covered by the metagenome sequencing data and how much was “unknown function”? Usually, only approximately 30-50% of the genes can be assigned functionally, thus leaving a large “black hole”. This has severe implications on the validity of determining similarity between GM15 and SPF.

3.) Selection of the GM15 members: It is not made clear, why exactly these 15 members of the GM15 were included and not others. Does this 15 member consortium cover most metagenome percentage/functionalities? Why not 16 or 14? Why was, for example, Akkermansia omitted, which is a known important gut bacterium involved in many beneficial aspects of host physiology?

4.) Standardization: Throughout the manuscript, the authors claim that use of GM15 would improve standardization in microbiome-related research. This is however not demonstrated clearly.

a) Most important, replication in another facility is lacking, ideally on a phenotypic trait that in SPF differs between these animal houses. This would clearly argue for an improved Standardization using GM15. This is especially important given that some other experimental parameters vary between facilities, e.g. food, routine procedures, bedding, housing, water and many more. AS the authors state “Indeed, the microbiota fluctuates a lot with diet and environment, so it is impossible to have

the exact same microbiota of SPF mice in two different facilities.”. This also applies to GM15 and the effects could be the same, larger or smaller compared to SPF.

b) 3 of the 15 strains did seem to not stably colonize throughout vertical transmission:

Lachnospiraceae bacterium MD335, Lachnospiraceae sp. MD329, and Lachnospiraceae sp. MD308. What does this mean in terms of long-term stability & reproducibility? Will it really be beneficial to always perform GM15 colonizations of GF mice and then the “real experiment” compared to having a continuous SPF colony?

5.) The phenotypic protection in the malnutrition model seems to be rather mild (10% higher body weight & length compared to SOPF, delta of ~1g or 0.6 cm). Although statistically significant, how biologically relevant is this delta?

6.) A comparison with other minimal microbiomes is lacking. The authors included ASF and OMM in the initial metagenome comparison, yet a functional comparison is lacking. This is especially important considering that much of the metagenomic functionality is unknown (see point #2). It would be important to include a comparison of the GM15 to GF mice or those colonized with OMM, ASF or another minimal microbiome, e.g. SIM (10.1016/j.celrep.2019.02.090)?

Overall, this study is highly interesting and relevant to the field, yet in its current form does not sufficiently support the central claims/findings. I am, however, confident that the authors are able to swiftly address the comments raised.

Reviewer #2 (Remarks to the Author):

In their publication the authors describe a new model of a new model of a standardize low-complexity microbiota in mice, named GM15. The authors show that the GM15 model can compensate some of the phenotype of the GF model such as breeding, growth, immune endocrinal and metabolic limitation while recapitulating many of the SOPF phenotypical features but without completely restore these features as show by the phenotype of the GM15 mice that is somewhere between the phenotype of GF or SOPF mice for most of the parameter asses by the authors. In the same way two of the main difference pointed out by the authors are the fact that the level IgE is elevated in the GM15 mice and a relative protection against to diet-induced stunting compare to SOPF mice. The authors conclude that GM15 is a novel controlled preclinical model phenotypically similar to SOPF.

The authors use a range of experimental assays to convincingly show that the phenotype of GM15 mice is closer to the one of SOPF mice than GF mice. However, in order to establish a new low-complexity microbiota model the manuscript suffers from a lack of comparison with already established and widely used models such as SPF and OligoMM (except for the number of microbiome functionalities covered) and some other issues need to be addressed. Detailed comments are below:

Major concerns:

1. The authors state that GM15 model offers new insight for gnotobiotic research as a complementary model of ASF and OligoMM. However, the authors don't offer a real comparison of these 3 models that could offer insight on the advantage or inconvenience of this model compared to the other. Furthermore, the majority of the manuscript aims to show that GM15 model phenotype is close to SOPF phenotype but the only pathogenic model used in the publication shows that GM15 mice present a different response to diet-induced stunting. The authors explain this difference by the possible action of a *Lactobacillus* strain. Both ASF and OligoMM models present *Lactobacillus* strain, so will they also present a protective effect against diet-induced stunting.
2. The data on colonization and stability of the GM15 consortium are very conventional but it is unclear if the founder mice and the different generations were housed in a same isolated environment. If it was the case, it will be interesting to have different sets of founder mice housed in different isolated environments and see if the GM15 consortium colonizes these mice in the same way as one of the first arguments for the GM15 consortium was that every strain was commercially available and thus easy to replicate in any facility.
3. The reason the authors use the diet-induced stunting model is unclear especially as the authors pursue further to explain the different phenotypes observed between GM15 and SOPF mice in this publication. If the point was to show an example of GM15 phenotype in a pathogenic model involving gut microbiota, it will have been interesting to have other examples in other models widely used. ASF and OligoMM models are widely used to study inflammatory bowel disease or obesity. Knowing the phenotype of GM15 mice in these models could be interesting especially since the authors report an elevated level of IgE that could lead to increased inflammation or unchanged levels of glucagon, insulin and leptin compared to GF mice.

Minor concerns:

1. In figure 1.A, the origin of the 4 mice used to define a normal SOPF microbiota is unclear. If these 4 mice were born and bred in the authors' facility, can we say that the microbiota used to define a normal SOPF microbiota is a widely representation of SOPF mice or just of SOPF mice present in the authors' facility?

2. In figure 1.B the authors are not very clear on how the different strain were choose for the GM15 consortium. For example, why the GM15 consortium contained 3 strain belonging to Lactobacillaceae and only 1 belonging to Tenerallaceae or Ruminococcaceae when both these phyla are more abundant in the microbiota of the SOPF mice in fig 1.A than the first one?

Reviewer #3 (Remarks to the Author):

The authors describe a comprehensive analysis to demonstrate how a gnotobiotic mouse model compares to both specific and opportunistic pathogen free mice and germ free mice.

General comments

The authors have conducted a range of tests to demonstrate the equivalence or not of this mouse model to SOPF. They have concentrated mainly on measures of immune system function which is perhaps not surprising given the link between microbiota and immune system.

Their major argument for the use of this model is to standardize preclinical studies in the host microbiome field. I am therefore slightly surprised given the range of tests they have undertaken on these mice that they did not use a pre-clinical test case to demonstrate that the results were similar to those of SOPF mice. Equally, in their discussion, they seem to be advising the use of this mouse as a model for allergy studies, which would suggest it would not make a good general model, thus creating a confusion for the reader on the ultimate use for this mouse.

The authors have used only NMR to assess metabolite profiles. NMR is excellent, but limited in its scope, whereas a combination of NMR and mass spectrometry would have allowed a much greater breadth of metabolites to be assessed and quantified.

Specific comments

Line 208: the authors state that they developed a strain specific qPCR assay to quantify the individual strains, but later in the article, around half of the strains cannot be quantified. How then can the authors be sure that the microbiome is as identified and stable as claimed? Can they also give more details on the “universal” primers they used (line 649), especially in relation to microbiome bacteria that may have been missed by this screening.

Line 230: did the GM15 mice age in a physiologically normal way, especially past 12 months old, which I believe still falls in the “middle age” range for this mouse type.

Line 245: how does this fold change compare with a mouse of the same strain with a standard microbiome

Line 260: the distribution of progeny per litter looks very different between GM15 and SOPF mice. This may be statistical chance, but it should be mentioned here, especially as there is also a decrease in the mean number of pups per mouse overall.

Line 322: does this observation of elevated IgE levels, combined with the reduced breeding capacity of SG15 mice not suggest to the authors that the gut diversity in SG15 mice is currently too minimal for them to be physiologically fully normal healthy mice?

Fig S4A has a half finished legend.

Line 364: the authors have specifically excluded sex dependent metabolites thus giving the reader no chance to see if there may be sex-dependent changes in these mice compared to SOPF mice. Please analyse and report these metabolites by individual sex/strain. Looking at other metabolites in this way would also be interesting.

Table S4: a big male to female difference in cholesteryl esters was reported in the GM15 mice but not shown in the other strains. TAGs may also be affected which may or may not indicate some differences in the bile acid and secondary bile acid production by this model. Was this explored?

Please also restructure the table to allow these results to be better compared by individual sex. Please highlight these sex dependent differences in the main text.

Line 420: please comment on the increased amounts of WAT and BAT in SOPF animals.

Line 432: I find this analysis and the conclusions drawn from it misleading. The proportional loss of brown fat is much greater in SOPF mice compared to GM15, and yet is not discussed and excluded from further analysis. The results of this further analysis which state that GM15 is within the normal phenotype of SOPF are thus highly misleading. Were the changes within a strain not assessed and compared?

Line 528: the manufacturer's instructions were not easy to find. Please reference a specific webpage or paper.

Line 718: if no technical replicates were assessed for these samples, please give details about the known technical error of these assays on your instrument when similar matrices are analysed.

Line 734: please give exact details of the modifications to this method to allow someone to replicate it as according to good scientific practice.

Line 750: when exactly was the DSS-d4 standard added to the samples. Why was there a need to calibrate the concentration?

Please give more details about how and when you prepared your QC samples for NMR.

Line 753: what was the final concentration of formic acid used?

Line 761-2: please give the technical reproducibility of your modified method, or link to a reference where you have published it.

Line 770: this section is a little unclear. Should this read “pooled organic phase”?

Line 794 onwards: please give full details here of the data preprocessing including any normalisation, missing values treatment and transformations applied. Was data pre-checked for any unusual results (batch effects, outliers, PCA or other visualisations etc) before PLS-Da was applied.

POINT BY POINT ANSWER TO REVIEWERS COMMENTS on manuscript NCOMMS-20-07272

We would like to thank the reviewers for their helpful comments and constructive criticisms that helped improving our manuscript. We have embarked on a very significant revision work and believe that we have now addressed the points raised during the first round of reviews. We think that our revised text and our new results have significantly improved the work and the manuscript.

Briefly, we have now significantly edited our manuscript and provide new experimental data demonstrating that (1) the GM15 model can be effectively established in another facility in a reproducible manner; (2) GM15 animals are phenotypically similar to SPF animals in another facility; (3) In contrast to SPF/SOPF animals, the GM15 model ensures a more reproducible animal growth dynamic across facilities. Finally, (4) we now include phenotypical comparisons between GM15 and Oligo-MM¹² gnotobiotic models at steady-state and we reveal that both models are phenotypically similar to SPF animals, yet each gnotobiotic model presents unique and specific phenotypical signatures which are of interest for future studies.

Please find below a point-by-point answer to the reviewers' comments (copied in bold below).

Reviewer #1 (Remarks to the Author):

The manuscript "A standardized gnotobiotic mouse model harboring a minimal 15-member mouse gut microbiota recapitulates SOPF phenotype" by Darnaud et al. describes the generation of the minimal microbiome GM15, which consists of 15 bacterial species, in an attempt to standardize microbiome-related experimental research. Mice colonized with GM15 phenotypically resemble SPF mice in some aspects. The GM15 mice were also less sensitive to protein malnutrition. The authors conclude that (i) the GM15 model is a more stable experimental model than SPF, which should improve reproducibility in microbiome experimentation, and that (ii) the GM15 model outperforms other minimal microbiomes, e.g. ASF or OMM. The topic of this study is of considerable interest. However, there are also significant concerns/limitations that should be addressed to substantiate these main findings:

1.) Four mice were used to generate the initial metagenome. How representative were those four animals (2 males, 2 females)? Were these littermates, from the same or different cages? How old?

We thank the reviewer for this remark and we have now added this information in our text (lines 130-140). Those four SOPF mice originate from a C57BL/6J SOPF colony from Charles River Lab (France). They were littermates but males and females were housed in different cages from weaning at 3 weeks old onwards, and feces were collected at 2 months old in our facility.

We also now provide in revised Fig. S1a, the distribution of bacterial abundance at the family level, represented by boxplots, for each SOPF individual animals (F1, F9, M1, M3). The p-value of the ANOVA parametric test was 0.95, indicating that there was no global significant

differences between samples. Then, all paired comparison by t test were not significant (ns), which indicates similar distributions between samples. In addition, the distribution of the log-normalized abundance for each bacterial family (in line) for each mouse (in column) was represented in a heatmap (revised Fig. S1b), which shows a very high similarity among different mice.

2.) In the KEGG Orthology analysis, how much of the genome was covered by the metagenome sequencing data and how much was “unknown function”? Usually, only approximately 30-50% of the genes can be assigned functionally, thus leaving a large “black hole”. This has severe implications on the validity of determining similarity between GM15 and SPF.

The ratio of assigned KEGG orthologs compared to the total number of coding genes was comparable between GM15 (44%), SOPF (47%), ASF8 (43%) and Oligo-MM¹² (46%). We have now added this information in our text lines 177, 200 and 202.

3.) Selection of the GM15 members: It is not made clear, why exactly these 15 members of the GM15 were included and not others. Does this 15 member consortium cover most metagenome percentage/functionalities? Why not 16 or 14? Why was, for example, Akkermansia omitted, which is a known important gut bacterium involved in many beneficial aspects of host physiology?

The design of the GM15 consortium was based on the most prevalent and representative bacteria detected by shotgun analysis from our SOPF mice founders, but also by the feasibility to isolate such bacteria from SOPF feces and caecum. For example, Akkermansia was not included in the consortium since Verrucomicrobiaceae was not identified as a prevalent and representative bacterial family by sequencing SOPF microbiota (Fig. 1a). In addition, 15 was an ideal community members number for qPCR monitoring using 48x48 microfluidic plates. Finally, *in silico* metagenome functional analysis confirmed that this minimal consortium was relatively well representative of a SOPF metagenome.

4.) Standardization: Throughout the manuscript, the authors claim that use of GM15 would improve standardization in microbiome-related research. This is however not demonstrated clearly.

We thank the reviewer for raising this point, it is indeed an important point and we believe our revised work now addresses this issue (see below).

a) Most important, replication in another facility is lacking, ideally on a phenotypic trait that in SPF differs between these animal houses. This would clearly argue for an improved Standardization using GM15. This is especially important given that some other experimental parameters vary between facilities, e.g. food, routine procedures, bedding, housing, water and many more. AS the authors state “Indeed, the microbiota fluctuates a lot with diet and environment, so it is impossible to have the exact same microbiota of SPF mice in two different facilities.”. This also applies to GM15 and the effects could be the same, larger or smaller compared to SPF.

We have embarked in a large revision work to re-establish and phenotype the GM15 model in another animal facility and compare them with SPF animals from this second facility. We have now generated a large set of new data reported in our revised manuscript in 2 new figures (Fig.7 and 8) and 3 new supplementary figures (Fig. S7, S8 and S9) and have added two additional results sections in the manuscript (lines 452-558). Our new results establish that (1)

the GM15 model is transferrable to another facility and that (2) GM15 phenotypical similarities to SPF animals are reproducible between facilities. Importantly, (3) the GM15 model ensures a more reproducible growth dynamics across facilities than SPF/SOPF animals (Fig. 8b). These new results support the notion that controlled minimal microbiota gnotobiotic models would ensure improved standardization in terms of microbiota composition and phenotypes at steady states across facilities.

b) 3 of the 15 strains did seem to not stably colonize throughout vertical transmission: Lachnospiraceae bacterium MD335, Lachnospiraceae sp. MD329, and Lachnospiraceae sp. MD308. What does this mean in terms of long-term stability & reproducibility? Will it really be beneficial to always perform GM15 colonizations of GF mice and then the “real experiment” compared to having a continuous SPF colony?

As we state in our manuscript (lines 242-246) the fact that we do not detect the three *Lachnospiraceae* strains with our qPCR assay may indeed stem from lack of colonization but could also rely the colonization by these strains of a specific niche that we have not yet sampled. A technical limit of detection of our assay is also to be envisaged. We cannot exclude these alternatives and future work will be required to try to identify these strains in GM15 animals. Of note, such limits of detection were also reported and confirmed in our analysis for 2 strains of the Oligo-MM¹² model. However, as far as standardization and reproducibility of phenotypes is concerned, our results establish that the GM15 community is stable *in vivo* during 9 filial generations (Fig. 2a – we have extended this reporting in the revised manuscript, initially we reported only 4 filial generations) and GM15 animal phenotypes are stable over two generations and across two facilities.

Concerning the last point, we do not advocate to systematically replace SPF/SOPF animal utilization by GM15 animals as gnotobiotic facilities won't replace SPF/SOPF facilities but we envisage that GM15 animals may be transferrable to and stable in individually ventilated cages from SOPF/SPF facilities, this will be the focus of a future work and publication. Yet at this stage, minimal microbiota models (such as GM15 or Oligo-MM¹²) provide an increased degree of standardization and reproducibility of preclinical studies and offer unique opportunities to study at the mechanistical level the symbiosis between a mammalian host and its microbiota members. We have clarified these points in our revised discussion and we now recommend to establish fresh GM15 animals on a regular basis to maintain standardization and reproducibility as the minimal microbiota of gnotobiotic models may also evolve as recently described with the Oligo-MM¹² model (Yilmaz et al. 2021) (see our revised text lines 605-609).

5.) The phenotypic protection in the malnutrition model seems to be rather mild (10% higher body weight & length compared to SOPF, delta of ~1g or 0.6 cm). Although statistically significant, how biologically relevant is this delta?

For non-expert of stunting the degree of protection of the GM15 and Oligo-MM¹² communities against stunting as compared to SPF/SOPF animals may seem mild, yet it is statistically significant and reproducible across facilities and biologically relevant. Indeed, please allow us to be anthropocentric here to illustrate how biologically significant is this phenotype: stunting is defined as reduced length-for-age by two standard deviations from the global median and according to WHO standards, the expected size (stature) of a 5-year-old boy is 110 cm (with 5 cm standard deviation) so a decrease of 2 standard deviations from the median is 100 cm in size, i.e. a 10% size decrease. So we believe that the effect reported in the manuscript of a 10% increase in weight and length in the gnotobiotic models vs SPF/SOPF animals upon chronic undernutrition is very significant biologically.

These results and models now offer the possibility to tackle the mechanistical roots of this phenotype and dig into the understanding of how the intestinal microbiota shapes its host response to poor nutrition.

6.) A comparison with other minimal microbiomes is lacking. The authors included ASF and OMM in the initial metagenome comparison, yet a functional comparison is lacking. This is especially important considering that much of the metagenomic functionality is unknown (see point #2). It would be important to include a comparison of the GM15 to GF mice or those colonized with OMM, ASF or another minimal microbiome, e.g. SIM (10.1016/j.celrep.2019.02.090)?

We thank the reviewer for this comment, we have now performed a side-by-side comparison of the GM15 and Oligo-MM¹² models. Our phenotypical comparison between the GM15 and Oligo-MM¹² models reveals that both gnotobiotic animals largely mimic SPF/SOPF animal phenotypes at steady state and both buffer diet-induced stunting. Interestingly each model also carries intrinsic phenotypical features which pave the way to study the underlying symbiotic mechanisms supporting these shared or unique phenotypes. These controlled minimal microbiota gnotobiotic models are unique tools to study these mechanisms in mice with an unprecedented microbial resolution. We have updated our manuscript accordingly (lines 472-558, 598-609).

Overall, this study is highly interesting and relevant to the field, yet in its current form does not sufficiently support the central claims/findings. I am, however, confident that the authors are able to swiftly address the comments raised.

We thank the reviewer for these final supportive words.

Reviewer #2 (Remarks to the Author):

In their publication the authors describe a new model of a new model of a standardize low-complexity microbiota in mice, named GM15. The authors show that the GM15 model can compensate some of the phenotype of the GF model such as breeding, growth, immune endocrinal and metabolic limitation while recapitulating many of the SOPF phenotypical features but without completely restore these features as show by the phenotype of the GM15 mice that is somewhere between the phenotype of GF or SOPF mice for most of the parameter asses by the authors. In the same way two of the main difference pointed out by the authors are the fact that the level IgE is elevated in the GM15 mice and a relative protection against to diet-induced stunting compare to SOPF mice. The authors conclude that GM15 is a novel controlled preclinical model phenotypically similar to SOPF.

The authors use a range of experimental assays to convincingly show that the phenotype of GM15 mice is closer to the one of SOPF mice than GF mice. However, in order to establish a new low-complexity microbiota model the manuscript suffer from a lack of comparison with

already establish and widely use model such as SPF and OligoMM (except for the number of microbiome functionalities covered) and some other issues need to be addressed.

We thank this reviewer for the comments and we now report a phenotypical comparison between GM15 and Oligo-MM¹² animals.

Detailed comments are below:

Major concerns:

1. The authors state that GM15 model offers new insight for gnotobiotic research as a complementary model of ASF and OligoMM. However, the authors don't offer a real comparison of these 3 models that could offer insight on the advantage or inconvenient of this model compare to the other. Furthermore, the majority of the manuscript aim to show that GM15 model phenotype is close to SOPF phenotype but the only pathogenic model use in the publication show that GM15 mice present a different response to diet-induced stunting. The authors explain this difference by the possible action of a Lactobacillus strain. Both ASF and OligoMM model present Lactobacillus strain, so will they also present a protective effect against diet-induced stunting.

We thank the reviewer for this comment and have addressed it by comparing the GM15 model to the Oligo-MM¹² model. Yet, we would like to emphasize that the aim of this study is to benchmark the GM15 model to SOPF/SPF mice. Of note, diet-induced stunting has been used in our study as a physiopathological model to highlight patterns that GM15 shares with SOPF/SPF mice, but also differences that could be correlated with minimal microbiome.

Our phenotypical comparison between the GM15 and Oligo-MM¹² models reveals that both gnotobiotic animals largely mimic SPF/SOPF animal phenotypes at steady state and both buffer diet-induced stunting. Interestingly each model also carries intrinsic phenotypical specificities which pave the way to study the underlying symbiotic mechanisms supporting these shared or unique phenotypes. These controlled minimal microbiota models are unique tools to study these mechanisms in mice with an unprecedented microbial resolution. We have updated our manuscript accordingly (lines 472-558, 598-609).

2. The data on colonization and stability of the GM15 consortium are very convention but it is unclear if the funder mice and the different generations were house in a same isolated environment. If it was the case, it will be interesting to have different set of funding mice house in different isolated environment and see if the GM15 consortium colonize these mice in the same way as one of the first argument for the GM15 consortium was that every strain was commercially available and thus easy to replicate in any facility.

We have generated new data for the revision of the manuscript and now show that it was possible to implement a reproducible GM15 colony in another facility by colonizing GF mice with the GM15 consortium (Fig. 7a, lines 454-472). Please note that every strain of the GM15 consortium will be available upon request to Bioaster under specific transfer and utilization conditions (for non-commercial purpose only).

3. The reason the authors use the diet-induced stunt model is unclear especially as the authors pursue further to explain the different phenotype observe between GM15 and SOPF mice in this publication. If the point was to show an example of GM15 phenotype in a pathogenic model involving gut microbiota, it will have been interesting to have others examples in others

models widely use. ASF and OligoMM model are widely use to study inflammatory bowel disease or obesity. Knowing the phenotype of GM15 mice in these models could be interesting especially since the authors report an elevated level of IgE that could lead to increase inflammation or unchanged level of glucagon, insulin and leptin compare to GF mice.

We believe that diet-induced stunting is a physiopathological context in which the contribution of the microbiota is now well established and recognized (Gordon and Leulier labs, Science papers, 2016). We have extensive expertise using this model and now extensive data using this model with GM15. We feel diet-induced stunting model is also an ideal set-up to start comparing GM15 to Oligo-MM¹² and our revised work now provide this comparison (and as indicated above we also compared both models at steady-state) (lines 477-517).

Again, the point of our work is not to benchmark our model with other existing gnotobiotic models in multiple physiopathological contexts, but to establish a controlled minimal mouse microbiota model which recapitulates most of the SOPF/SPF mice phenotype at steady state with a level of phenotypical characterization that has not yet been reported for any minimal gnotobiotic model so far. However, using GM15 and comparing it to Oligo-MM¹² in other disease-related context (beyond diet-induced stunting) is an important avenue of research and will be the focus of future projects which are at this stage beyond the scope of this first publication.

Finally, we reviewed extensively the elevated IgE and IL17a signatures detected in GM15 animals and we are now tuning down our conclusion as we observed similar levels over time between GM15 and SOPF mice in facility 1 (new Fig. 4m and 4o, and S4f), as well as in the GM15 model established in facility 2 (new Fig. S9a-c) (lines 339-355, 521-536).

Minor concerns:

1. In figure 1.A, the origin of the 4 mice use to define a normal SOPF microbiota is unclear. If these 4 mice were born and breed in the authors facility, can we say that the microbiota use to define a normal SOPF microbiota is widely representation of SOPF mice or just of SOPF mice present in the authors facility?

The 4 SOPF mice used in Figure 1a were obtained from a SOPF colony from Charles River Lab (France) and then housed in our animal facility. They were littermates but males and females were housed in different cages from weaning at 3 weeks old onwards, and feces were collected at 2 months old in our facility (lines 130-140). Thus, these mice are well representative of a normal SOPF microbiota sold by a major mice manufacturer. Additional data generated for the revision of the manuscript show that 13 strains out of the 15 strains of the GM15 model are also detected in completely unrelated SPF mice, which were born and bred in an independent and distant animal facility.

2. In figure 1.B the authors are not very clear on how the different strain were choose for the GM15 consortium. For example, why the GM15 consortium contained 3 strain belonging to Lactobacillaceae and only 1 belonging to Tenerallaceae or Ruminococcaceae when both these phyla are more abundant in the microbiota of the SOPF mice in fig 1.A than the first one?

The design of the GM15 consortium was indeed based on the most prevalent and representative bacteria detected by shotgun analysis from our SOPF mice, but also by the feasibility to isolate such bacteria from SOPF feces and caecum. For example, we faced difficulties to isolate a strain belonging to Deferribacteraceae. However, if we consider bacterial

load in GM15 mice (Fig. 2e), *Parabacteroides goldsteinii* MD072 (Tannerellaceae) contribute relatively more than the 3 lactobacilli together, and the 3 lactobacilli cumulative loads reach a similar titers than *Anaerotruncus colihominis* (Ruminococcaceae).

Reviewer #3 (Remarks to the Author):

The authors describe a comprehensive analysis to demonstrate how a gnotobiotic mouse model compares to both specific and opportunistic pathogen free mice and germ free mice.

General comments

The authors have conducted a range of tests to demonstrate the equivalence or not of this mouse model to SOPF. They have concentrated mainly on measures of immune system function which is perhaps not surprising given the link between microbiota and immune system.

Their major argument for the use of this model is to standardize preclinical studies in the host microbiome field. I am therefore slightly surprised given the range of tests they have undertaken on these mice that they did not use a pre-clinical test case to demonstrate that the results were similar to those of SOPF mice. Equally, in their discussion, they seem to be advising the use of this mouse as a model for allergy studies, which would suggest it would not make a good general model, thus creating a confusion for the reader on the ultimate use for this mouse.

We thank this reviewer for this comment and we have addressed these points in our revised manuscript with both revised wording and additional work.

We now report how the GM15 model compare to SPF/SOPF animals in two distinct facilities at steady state and in the preclinical test case of diet-induced stunting. In addition, we now report both at steady state and in the test case the comparison of the GM15 model with another minimal microbiota gnotobiotic model, Oligo-MM¹².

The point of our work is not to benchmark our model with other existing gnotobiotic models in multiple physiopathological contexts, but to establish a controlled minimal mouse microbiota model which recapitulates most of the SOPF/SPF mice phenotype at steady state with a level of phenotypical characterization that has not yet been reported for any minimal gnotobiotic model so far.

However, our phenotypical comparison between the GM15 and Oligo-MM¹² models reveals that both gnotobiotic animals largely mimic SPF/SOPF animal phenotypes at steady state and both buffer diet-induced stunting. Interestingly, each model also carries intrinsic phenotypical specificities which pave the way to study the underlying symbiotic mechanisms supporting these shared or unique phenotypes. These controlled minimal microbiota models are unique tools to study these mechanisms in mice with an unprecedented microbial resolution. We have updated our manuscript accordingly (lines 472-558, 598-609). Using GM15 and comparing it to Oligo-MM¹² in other disease-related context (beyond diet-induced stunting) is an important avenue of research and will be the focus of future projects which are at this stage beyond the scope of this first publication.

The authors have used only NMR to assess metabolite profiles. NMR is excellent, but limited in its scope, whereas a combination of NMR and mass spectrometry would have allowed a much greater breadth of metabolites to be assessed and quantified.

We thank the reviewer for this comment and we agree that a combination of NMR and mass spectrometry would have been more resolutive. However, our aim was not to embark in a deep metabolic phenotyping of the animals but rather define the global phenotypic landscape of the GM15 animals and compare it to GF, SOPF/SPF and Oligo-MM¹² animals.

Specific comments

Line 208: the authors state that they developed a strain specific qPCR assay to quantify the individual strains, but later in the article, around half of the strains cannot be quantified. How then can the authors be sure that the microbiome is as identified and stable as claimed? Can they also give more details on the “universal” primers they used (line 649), especially in relation to microbiome bacteria that may have been missed by this screening.

As we state in our manuscript (lines 242-246) the fact that we do not detect the three *Lachnospiraceae* strains out of the 15 strains of the GM15 model with our qPCR assay may indeed stem from lack of colonization but could also rely on the colonization by these strains of a specific niche that we have not yet sampled. A technical limit of detection of our assay is also to be envisaged. Indeed, like any technology, qPCR carries intrinsic limits of detection and 3/15 strains of the GM15 model are indeed never detected in GM15 animals by qPCR assay despite being inoculated in a viable state to the animals and detected in SOPF animals. Of note, such limits of detection were also reported and confirmed in our analysis for 2 strains of the Oligo-MM¹² model. Future work will be required to try to identify these strains in GM15 animals. However, as far as standardization and reproducibility of phenotypes is concerned, our results establish that the GM15 community (at least the 12 strains detected by qPCR) is stable *in vivo* during 9 filial generations (Fig. 2a – we have extended this reporting in the revised manuscript, initially we reported only 4 filial generations) and GM15 animal phenotypes are stable over two generations and across two facilities.

We believe that the other missing strains in specific mice or conditions stem from their low abundance in the consortium and therefore sometimes they fall below the limit of detection in fecal DNA extracts. However, these strains are viable upon inoculation and coprophagy in mice suggests that littermates may share these low-abundance bacterial strains even if they are not detected by qPCR. Hence, these strains were detected by qPCRs in older mice or in filial generations suggesting that they were present at younger age even if they were not detected at that time.

We used the “universal” primers UniF/R targeting highly conserved sequences within the *E. coli* 16S rRNA gene (1542 nucleotides) and known as a generic primer set to amplify the genes encoding 16S rRNA from all bacteria (Packey et al. 2013). UniF is complementary to nucleotides 1047 through 1067, while UniR is complementary to nucleotides 1174 through 1194, generating amplicons of 147 base pairs (Table S2).

Line 230: did the GM15 mice age in a physiologically normal way, especially past 12 months old, which I believe still falls in the “middle age” range for this mouse type.

In this study, we did not primarily aimed at assessing the impact of aging on the GM15 microbiota. However, the follow-up of 9 GM15 mice during 12 months did not show any

abnormal clinical signs. We do not have any data past 12 months, mainly because we do not use older mice in our studies. This would indeed be interesting to assess GM15 mice for aging studies but it is currently beyond the scope of this work.

Line 245: how does this fold change compare with a mouse of the same strain with a standard microbiome

In our revised work we have compared the titers of each strain individually in GM15 and SOPF/SPF animals from both facilities. Beside *Clostridium* sp. MD294 and *Clostridium* sp. MD300 which are less well colonizing GM15 animals than SPF/SOPF animals in both facilities, there was a clear general tendency of a reproducible colonization illustrated by more similar fecal titers in GM15 animals as compared to their titers in SPF/SOPF animals between facilities (Fig. 7b and S7a). These results demonstrate that the GM15 model can be effectively transferred and established in a reproducible manner in different gnotobiotic facilities. We have added this new piece of data to our revised manuscript (lines 467-472).

Line 260: the distribution of progeny per litter looks very different between GM15 and SOPF mice. This may be statistical chance, but it should be mentioned here, especially as there is also a decrease in the mean number of pups per mouse overall.

Indeed, the distribution of progeny per litter in SOPF mice ranges from 3 to 10 pups but is mostly centered on 7 or 8 pups. On the other part, the distribution in GM15 mice ranges more evenly from 3 to 9 pups, with a short majority of 7 pups per litter. This information was added in lines 276-279.

Line 322: does this observation of elevated IgE levels, combined with the reduced breeding capacity of SG15 mice not suggest to the authors that the gut diversity in SG15 mice is currently too minimal for them to be physiologically fully normal healthy mice?

We have reviewed extensively the elevated IgE signature detected in GM15 animals by testing IgE levels in additional GM15 animals from facility 1 and from another facility. We have also quantified IgE using two alternative methods. We are now tuning down our conclusions on this matter as we observed similar IgE levels over time between GM15 and SOPF mice in facility 1 (new Fig. 4m and S4f), as well as in the GM15 model established in facility 2 (new Fig. S9a and S9b). In addition, apart from 3 outliers in facility 1, IL-17a levels expressed in GM15 mice were also similar to SOPF/SPF mice in both facilities (Fig. 4n and 4o, and S9c) (lines 339-355, 521-536).

Fig S4A has a half finished legend.

The legend has been updated.

Line 364: the authors have specifically excluded sex dependent metabolites thus giving the reader no chance to see if there may be sex-dependent changes in these mice compared to SOPF mice. Please analyse and report these metabolites by individual sex/strain. Looking at other metabolites in this way would also be interesting.

We have reformulated lines 366-369 for more clarity: "Using this technology, we analyzed the metabolic profile of plasma samples from GF, GM15 and SOPF mice and were able to quantify a total of 57 polar metabolites and 5 non-polar metabolites. It could be seen that only few of them were significantly affected by sex (Additional file 7: Table S3)." The new Table S3 summarizes all the metabolites concentrations for each animal group and each sex, for both animal facilities after reviewing annotation based on updated knowledge (Bliziotis et al. 2020,

<https://doi.org/10.1007/s11306-020-01686-y>; Guo et al. 2016,
<https://doi.org/10.1021/acs.jproteome.6b00179>; Nagana Gowda et al. 2015;
<https://doi.org/10.1021/ac503651e>; Wijeyesekera et al. 2012,
<https://doi.org/10.1021/pr2010154>) and quantification with new version of Chenomx NMR suite 8.6.

Table S4: a big male to female difference in cholesteryl esters was reported in the GM15 mice but not shown in the other strains. TAGs may also be affected which may or may not indicate some differences in the bile acid and secondary bile acid production by this model. Was this explored?

Actually, there was no sex effect on cholesteryl esters regardless of the mouse strain, and male to female difference (fold change > 1.3 and p-value < 0.05) was only observed in triacylglyceride in GF mice from facility 1 and in SPF from facility 2 (revised Table S3). The analysis of bile acids and secondary bile acids, was beyond the scope of plasma metabolic phenotyping. In the same time, unfortunately, in our NMR data, we were unable to explore bile acids or secondary bile acids. These molecules were below our detection limit and a LC-MS approach might be necessary.

Please also restructure the table to allow these results to be better compared by individual sex. Please highlight these sex dependent differences in the main text.

The intention of the publication was to give a global metabolic overview that is why we used multivariate analysis to explore the data. Nevertheless, we have now replaced the previous Tables S3 and S4 by a new one (revised Table S3) which contains the raw data for each mice in the different conditions tested.

Line 420: please comment on the increased amounts of WAT and BAT in SOPF animals.

We respectfully disagree with reviewer #3. There was no statistically significant increase in any of the adipose depots in SOPF mice, as shown by ANOVA2. However, WAT mass was correlated to body weight and to the weight of other organs, and was thus included in the Principal Coordinate Analysis.

Line 432: I find this analysis and the conclusions drawn from it misleading. The proportional loss of brown fat is much greater in SOPF mice compared to GM15, and yet is not discussed and excluded from further analysis. The results of this further analysis which state that GM15 is within the normal phenotype of SOPF are thus highly misleading. Were the changes within a strain not assessed and compared?

We respectfully disagree with reviewer #3. While the mean of groups is different, there was no statistical difference between any of the groups regarding BAT, as tested by ANOVA 2 (effect of microbiota: P = 0.60). When we ran a correlation matrix, BAT mass was not correlated to any of the parameters, and was thus left out of further analysis.

Line 528: the manufacturer's instructions were not easy to find. Please reference a specific webpage or paper.

Reference 73 was added in new line 653.

Line 718: if no technical replicates were assessed for these samples, please give details about the known technical error of these assays on your instrument when similar matrices are analysed.

The following details were added to the manuscript in lines 953-956. The known intra technical error (%CV) per assay was 7% for Insulin, 8% for Glucagon, 6% for Leptin, 15% for Ghrelin, 10% for IgA, 6% for IgG1, 8% for IgG2a, 7% for IgG2b, 5% for IgG3, 5% for IgM, 2% for IL-1b, 3% for IL-10, 2% for IL-12p70, 3% for IL-17a, 3% for IL-17f, and 3% for IL-22.

Line 734: please give exact details of the modifications to this method to allow someone to replicate it as according to good scientific practice.

Actually, all details were described in the chapter Polar metabolites preparation but we agree that adding the overview of the protocol above this chapter might be confusing. So we moved up and reorganized the chapter "Polar metabolites preparation" under lines 969 and 1014.

Line 750: when exactly was the DSS-d4 standard added to the samples. Why was there a need to calibrate the concentration?

We thank the reviewer for this question as this allowed us to correct an error in our manuscript since the standard solution used was DSS-d6 instead of DSS-d4. We modified the text (lines 1003-1012) by "Vortexed 10 kDa-filtered plasma samples (about 135 μ L) were transferred in a 0.5 mL 96-well plate (Agilent) and mixed on thermoshaker Eppendorf (1.5 min, 10°C, 650 rpm) with 45 μ L of internal standard solution DSS-d6 (1.54 mM), which also contains the pH-reference standard DFTMP (4 mM). The internal standard solution DSS-d6 was prepared with phosphate buffer (0.6 M, 60:40 v/v H₂O:D₂O, pH=7.4). Finally, 155 μ L of the resulting sample solutions were transferred in 3 mm SampleJet NMR tubes. Since the DSS-d6 solution might be unstable during long time storage (hygroscopic properties, trimethylsilyl fragment degradation etc.), the DSS-d6 concentration was calibrated by ¹H NMR using sodium succinate dibasic hexahydrate standard solution, a compound which have better stability properties, in order to guarantee data accuracy."

Please give more details about how and when you prepared your QC samples for NMR.

We added to the text (lines 1012-1014): "This protocol was systematically applied including for blank samples, which were prepared by replacing the plasma by Milli-Q water and for quality controls (n=7), which were prepared using commercially available mouse plasma." QC data are available in revised Table S3.

Line 753: what was the final concentration of formic acid used?

We modified the text "(2.1% in milliQ water)" by "(0.1% final concentration)" (line 983).

Line 761-2: please give the technical reproducibility of your modified method, or link to a reference where you have published it.

This information is now mentioned in lines 993-1002. We compared the supplier's washing protocol of 10 kDa filter (VWR) using centrifugation after one wash with sodium hydroxyde 0.05N or ethanol 70% followed by four washes with milli-Q water, with our ultrasonication approach performed 5 times in milli-Q water. We observed that glycerol was completely discarded in the ultrasonicated tubes (N = 5; average = under detection limit), while it was still present after centrifugation following washes with either sodium hydroxyde (N = 5; average = 0.02 mM; standard deviation = 0.02 mM), or ethanol (N = 5; average = 0.05 mM; standard deviation = 0.01 mM), or 5 control washes with milli-Q water (N = 5; average = 0.02 mM; standard deviation = 0.01 mM). In addition, we observed that the use of ethanol contributed to contamination since traces of ethanol remain. Finally, regarding the mechanical properties of the filters, we also observed that after 5 washes and centrifugations at 10000xg, filters collapsed during plasma filtration.

Line 770: this section is a little unclear. Should this read “pooled organic phase”?

Indeed, the text has been corrected as “pooled organic phase” (line 1022).

Line 794 onwards: please give full details here of the data preprocessing including any normalisation, missing values treatment and transformations applied. Was data pre-checked for any unusual results (batch effects, outliers, PCA or other visualisations etc) before PLS-Da was applied.

First of all, we systematically used internal standards throughout all batches analyzed by NMR that allow normalisation of analytical batch effects. In addition, analytical batch effects were pre-checked using PCA as unsupervised data analysis prior further bioinformatics analysis.

Then, potential biological effects due to mice' Gender, Generation, Age and Group (GF, GM15, SOPF) were also systematically checked. Here, we particularly assessed the relative contribution of each effect to our data after PCA analysis. The process consisted in assessing the statistical significance of each effect in the PCA coordinates (PC1 and PC2 score plots) by measuring the Mahalanobis distance between each group and the associated significance using a Fisher test. The strongest effect was Group effect, followed by Age, Generation and Gender. This was confirmed by supplementary multivariate supervised analyses (PLS-Da, figure below): Plot of the variance explained by the effect vs. the cross-validation coefficient of determination Q²_Y using a linear model, where Q²_Y represents the statistical significance assessed using permutation testing (with p-value < 0.001).

As we were mainly interested in the Group effect that was the most significant, we decided to not apply any data correction on the other effects.

Value under the limit of detection, when the signal to noise is below 3, were replaced by 0 at the NMR data quantification step with Chenomx NMR suite 8.6. For example, the dimethyl sulfone was detected in GM15 and SOPF mice, but not in GF mice. Thus, no values were excluded during the statistical analysis, and there was no missing values treatment and transformations applied.

REVIEWER COMMENTS

Reviewer #1 (Remarks to the Author):

The authors took a great effort with an extensive and careful revision. All my previous concerns have been addressed in full. I congratulate the authors on this manuscript and look forward to see this important piece of work published.

Reviewer #2 (Remarks to the Author):

The authors submitted a satisfying revision where they answered my concerns. I have no more comments and I think the paper is ready for publication

Reviewer #3 (Remarks to the Author):

The authors have made substantial revisions to their manuscript which have strengthened their claims. I am broadly satisfied and have only a few very minor points.

NMR is limited in the overview it can give of an animal's metabolome, but given the extensive additional experiments the reviewers have performed to show the stability of the phenotype, I am prepared to accept their suggestion that more extensive metabolomics can be conducted in a future study.

1) Brief details on the universal primer should be included in the methods section.

2) Fig S4 The figure legend states a much larger number of original mice than used in the final IgE ELISA assay. What are the reasons for this. There still appears to be a trend of increased IgE in these animals, even if not powered sufficiently to reach statistical significance.

3) Line 420: The authors state that there is no statistical difference between WAT and BAT in SOPF animals, but we see that in Fig 6 and S9 there is a statistical difference between GM15 and SPF visceral WAT under BD conditions and there are trends in other fat layers e.g. epididymal WAT. In some cases this emphasizes the fat loss that occurs in the SOPF animals between BD and DD diets even if they do not reach a level of statistical significance. Given that you have observed a change in

WAT levels in the female mice in the second study, I would suggest a line referencing these trends in the males are warranted.

I am also surprised given the very different fat coverage of male and female mice, that only male mice are pictured here, especially since authors mention that the differences were more significant in female mice.

4) Line 794 onwards: please add the data preprocessing details to the methods.

5) Fig 6g and line 438 – is a PCA plot shown, or a principal coordinate analysis plot? The labelling on the axes suggests the latter but it states PCA in both the text and the figure legend.

POINT BY POINT ANSWER TO REVIEWERS COMMENTS on manuscript NCOMMS-20-07272

We would like to thank the reviewers for their positive comments on our revised work.

Please find below a point-by-point answer to the reviewers' comments (copied in bold below).

Reviewer #3 (Remarks to the Author):

The authors have made substantial revisions to their manuscript which have strengthened their claims. I am broadly satisfied and have only a few very minor points.

NMR is limited in the overview it can give of an animal's metabolome, but given the extensive additional experiments the reviewers have performed to show the stability of the phenotype, I am prepared to accept their suggestion that more extensive metabolomics can be conducted in a future study.

We thank this reviewer for this comment and we have addressed the following points in our revised manuscript.

1) Brief details on the universal primer should be included in the methods section.

Addition has been made in lines 872-876.

2) Fig S4 The figure legend states a much larger number of original mice than used in the final IgE ELISA assay. What are the reasons for this. There still appears to be a trend of increased IgE in these animals, even if not powered sufficiently to reach statistical significance.

The reason for the difference of number of mice is that original mice (N~20, generations F1 and F2) were females and males of the phenotyping study (no sex effect), while additional mice (N=10, generations F5 and F6) used in the final IgE assay (Supplementary Fig. 4f) were control males fed with breeding diet (BD) in the stunting study. IgE ELISA raw data support that there is no difference between GM15 and SOPF males (GM15: 117.3; 149.9; 117.3; 117.3; 53.7; 72.5; 41.3; 53.7; 0.2; 35.2 ng/mL, and SOPF: 110.8; 78.8; 35.2; 35.2; 47.5; 17.2; 78.8; 549; 41.3; 98 ng/mL).

Legends of Fig. 4 (lines 1484-1485) and Supplementary Fig. 4 (lines 59-67 of the Supplementary Information file) have been modified for more clarity on these additional mice.

3) Line 420: The authors state that there is no statistical difference between WAT and BAT in SOPF animals, but we see that in Fig 6 and S9 there is a statistical difference between GM15 and SPF visceral WAT under BD conditions and there are trends in other fat layers e.g. epididymal WAT. In some cases this emphasizes the fat loss that occurs in the SOPF animals between BD and DD diets even if they do not reach a level of statistical significance. Given

that you have observed a change in WAT levels in the female mice in the second study, I would suggest a line referencing these trends in the males are warranted.

We thank the reviewers for this comment. We amended the result section accordingly (lines 486-490):

“In GM15 males, the WAT was significantly heavier as compared to SPF and Oligo-MM¹² males. This observation was different compared to the results from facility 1, where GM15 males’ visceral WAT was lighter compared to SOPF males on breeding diet (Supplementary Fig. 6b). We observed no difference regarding the BAT among male groups, both in facility 1 and facility 2.”

I am also surprised given the very different fat coverage of male and female mice, that only male mice are pictured here, especially since authors mention that the differences were more significant in female mice.

We believe the referee mentions Supplementary Fig. 8 results. Based on our previous experience with juvenile chronic undernutrition model, stunting is more prominent in male mice that experience the growth spurt after weaning, hence in the context of diet-induced stunting we only focused our studies on males in both facilities (Fig. 6a-f, Supplementary Fig. 6a-h, Fig. 8e-g, Supplementary Fig. 8b). However, our work describes macroscopic and tissue growth phenotypes on the breeding diet for both sexes (Fig. 3e-h, Fig. 8a-g, Supplementary Fig. 8a-b).

4) Line 794 onwards: please add the data preprocessing details to the methods.

Addition has been made in lines 1053-1060.

5) Fig 6g and line 438 – is a PCA plot shown, or a principal coordinate analysis plot? The labelling on the axes suggests the latter but it states PCA in both the text and the figure legend.

We thank the reviewer for raising this error and we corrected the text with “Principal Coordinates Analysis (PCoA)” in lines 433, 434, 436 and 1514.